# Talking Heads: Understanding Inter-layer Communication in Transformer Language Models

**Jack Merullo**
Department of Computer Science
Brown University
jack_merullo@brown.edu

**Carsten Eickhoff**
School of Medicine
University of Tübingen
carsten.eickhoff@uni-tuebingen.de

**Ellie Pavlick**
Department of Computer Science
Brown University
ellie_pavlick@brown.edu

## Abstract

Although it is known that transformer language models (LMs) pass features from early layers to later layers, it is not well understood how this information is represented and routed by the model. We analyze a mechanism used in two LMs to selectively inhibit items in a context in one task, and find that it underlies a commonly used abstraction across many context-retrieval behaviors. Specifically, we find that models write into low-rank subspaces of the residual stream to represent features which are then read out by later layers, forming low-rank **communication channels** [Elhage et al., 2021] between layers. A particular 3D subspace in model activations in GPT-2 can be traversed to positionally index items in lists, and we show that this mechanism can explain an otherwise arbitrary-seeming sensitivity of the model to the order of items in the prompt. That is, the model has trouble copying the correct information from context when many items "crowd" this limited space. By decomposing attention heads with the Singular Value Decomposition (SVD), we find that previously described interactions between heads separated by one or more layers can be predicted via analysis of their weight matrices alone. We show that it is possible to manipulate the internal model representations as well as edit model weights based on the mechanism we discover in order to significantly improve performance on our synthetic Laundry List task, which requires recall from a list, often improving task accuracy by over 20%. Our analysis reveals a surprisingly intricate interpretable structure learned from language model pretraining, and helps us understand why sophisticated LMs sometimes fail in simple domains, facilitating future analysis of more complex behaviors.[1]

## 1   Introduction

Despite the impressive capabilities of LMs, they often suffer from seemingly arbitrary sensitivities to prompts. These failure cases are particularly troubling because they are not systematic; it is very difficult to predict when, for example, the order of information seemingly randomly causes a model to fail [Pezeshkpour and Hruschka, 2023, Liu et al., 2024, Li and Gao, 2024, Zheng et al., 2024, Zhou et al., 2023], or the format of a prompt hurts performance [Liu et al., 2023, Sclar et al.,

---

[1] https://github.com/jmerullo/talking_heads.git

2023, Zhao et al., 2021, Lu et al., 2022, Webson and Pavlick, 2022]. As LLMs become increasingly ubiquitous, we will require more principled ways of anticipating and remedying unstable or unwanted behaviors [Yu et al., 2024, Yong et al., 2023]. Understanding the mechanisms in play within LLMs, and connecting those mechanisms to behavior, could enable such principled approaches.

One aim of interpretability research is to explain model behaviors, so is it possible to explain why some particular failure exists? In this paper, we consider a simple *laundry list task* that exhibits one such undesirable instability. Specifically: Transformer language models (LMs) struggle to reliably recall items from a list as the length of the list increases, and performance can vary wildly depending on the position of the item in the list that is being recalled (Figure 1). This instability is not obvious from the model architecture itself–i.e., unlike their predecessors Elman [1991], Transformers [Vaswani et al., 2017] can use attention to recall freely from anywhere in context. Thus, we use this task as a case study in order to connect the low-level *emergent mechanisms* which are encoded during LM pretraining to observable behavior, and illustrate as a proof of concept that a precise mechanistic understanding of LMs can be used to explicate and, perhaps, remedy model performance in practice.

Specifically, by building on recent work in circuit analysis [Elhage et al., 2021, Wang et al., 2022, Goldowsky-Dill et al., 2023, Quirke and Barez, 2024, Merullo et al., 2024, Hanna et al., 2023], we demonstrate how a Transformer LM (GPT2 small, Pythia 160m) passes information from early layers to later ones using low-rank subspaces. These *communication channels* are proposed in Elhage et al. [2021], but understanding their implementation in weights is an open challenge [Makelov et al., 2024]. This conflicts with circuit analysis, which supports the interpretation of specialization and communication between specific transformer components. Elhage et al. [2021] introduce a score for measuring how much weight matrices in a transformer communicate, but outside of toy models, it has been difficult to interpret the results [Singh et al., 2024]. We build on this work by proposing a method using the Singular Value Decomposition to find these channels by decomposing one matrix into all of its component signals, and find it is much more interpretable than the non-decomposed variant.We focus on two examples of such channels (inhibition and duplicate detection, §2), and find that they are are very low-rank (1 or 2 dimensions), easily interpretable, and causally important for specific model behaviors. We also show this method can be used to perform model editing at training time (Appendix H.2), and provide some encouraging early evidence we can perform circuit discovery without running models (Appendix I). Specifically, our contributions are as follows:

- We explore a fundamental question in interpretability on how information passed from layer to layer in an LM is represented internally. We find "communication channels" [Elhage et al., 2021] encoded in the weights that connect attention heads separated by several layers.

- We propose a simple extension to previous weight-based methods that more effectively isolates low-rank signals passed through such channels.

- We show that low-rank communication mechanism plays a role in prompt sensitivity on an item recall task that otherwise seems idiosyncratic, and that intervention in the mechanism can be used to affect downstream behavior to improve performance.

Our findings indicate more broadly that LMs are capable of learning intricately structured representations from self-supervised pretraining without inductive biases. This may have important implications for the emergence of abstract and content-independent operations, and for developing methods for steering and understanding these models (see Discussion, §7).

## 2  Background

Throughout our work, we consider decoder-only transformer language models and primarily use GPT2-Small. The modern attention mechanism used by these models Vaswani et al. [2017] uses multiheaded attention, where Query and Keys control *which tokens* from earlier in context are attended to and Value and Output matrices control *what information* is moved from these tokens. An important abstraction we use in our work relies on rewriting these matrices as the products QK (Query*Key) and OV (Output*Value)[2]. For an individual head, these matrices are themselves low-rank compared to the embedding dimension of the network (due to down-projecting the input e.g., they have size

---

[2]Following convention, we refer to this matrix as OV, but the Transformer Lens library implements right-hand matrix multiplication so we actually use VO. This does not effect our results

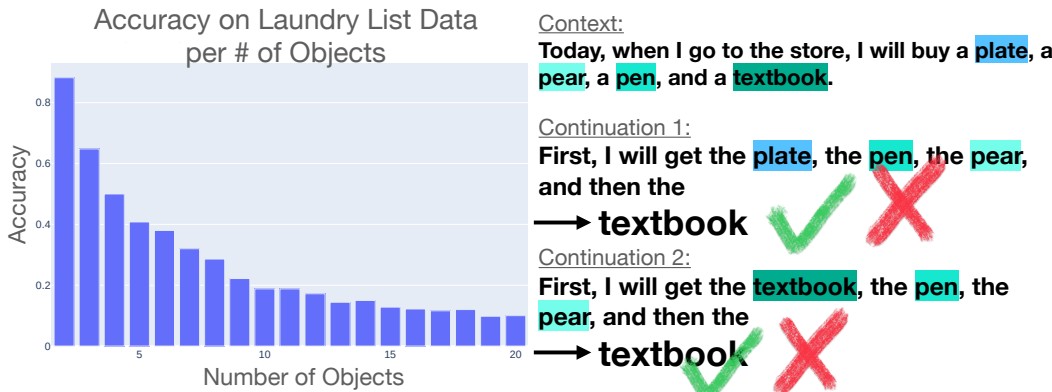

Figure 1: Language models are often sensitive to arbitrary changes in a prompt, for example the order in which objects are listed (right). This problem is more pronounced as the number of objects increases (left) even though it is not obvious where the issue stems from in the model. We broadly explore how information is routed through a model and focus on a mechanism that is in part responsible for this (in)ability.

768x64 dimensions in GPT-2). This is a useful property as we are motivated by looking for subspaces that are written into/read from by these matrices and this reduces the search space.

In order to ground our findings about inter-layer communication to real model behaviors, we focus on attention head interactions which we already understand and work backwards to determine how they communicate. Recent works in circuit analysis provide detailed explanations of how different model components interact on controlled datasets. In particular, we make use of the Indirect Object Identification (IOI) circuit discovered in Wang et al. [2022]. We use GPT2-Small, to study three specific types of interactions between heads: cases where heads write information that is used by the keys, queries, or values of later heads.

When we refer to an attention head as 3.0 or 7.9, this means layer 3, head 0 or layer 7 head 9.

**Three Types of Composition** In attention heads, there are three ways that earlier heads can contribute to the processing done downstream. In all cases, information is written into the residual stream by the OV matrix of an earlier head, and read back out by either the Query, Key, or Value matrices of a later head. These concepts are introduced in Elhage et al. [2021]. We also provide an example of each composition type that we examine further in Section 3. We look for communication channels in one of each type of composition. These are previous token to induction head composition (key) [Olsson et al., 2022, Singh et al., 2024, Reddy, 2023], duplicate token to inhibition head composition (value), and inhibition to mover head composition (query) [Wang et al., 2022]. The variation in the way these heads communicate only changes how we calculate the composition score [Elhage et al., 2021] and individual implementation of the communication, but we do not make claims about how these types of composition differ from each other in more meaningful ways.

**The Inhibition-Mover Subcircuit** We build on work from IOI [Wang et al., 2022] which documents a circuit that appears in multiple tasks [Merullo et al., 2024]. This circuit includes mover heads, which copy tokens from context to the output, and inhibition heads which (optionally) block the mover heads' attention to certain tokens and thus prevent certain tokens from being copied. Inhibition heads are known to receive signals from duplicate token head value vectors which help inform which tokens to inhibit. In GPT2-Small, the known inhibition heads are 7.3, 7.9, 8,6, and 8.10 and we consider their communication to the mover head 9.9[3]. This is the example we use for query composition experiments. In Section 5 we explore this circuit's role in problems of prompt sensitivity and a learned structure to control the indexing of tokens in the context window.

---

[3]for simplicity we only consider this mover head, but we do not find the choice matters much.

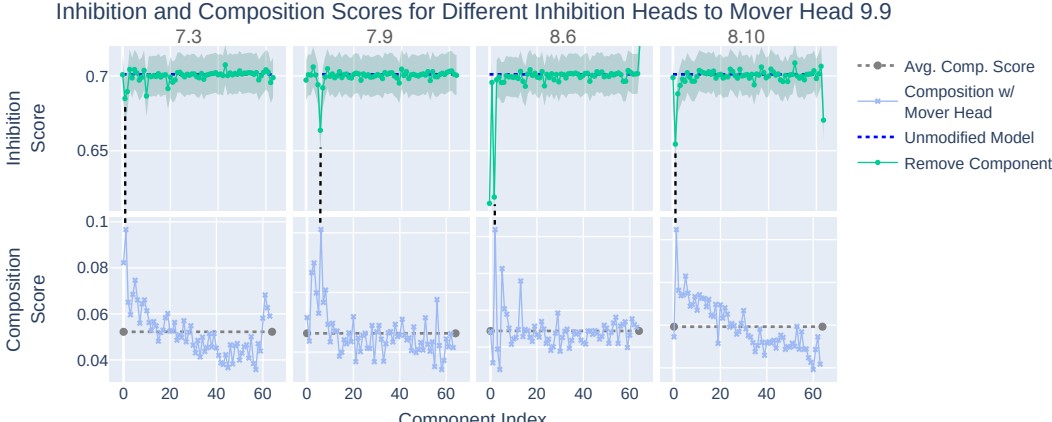

Figure 2: Showing the relationship between the composition score (weight-based, bottom) and inhibition score (data-based, top) between various inhibition head components and mover head 9.9 for the IOI task. The inhibition of each inhibition head is generally highly concentrated in one or two components of the matrix, removing it causes a large drop in the later mover head's ability to downweight one of the names. We therefore show that we can use the composition score when considering decomposed matrices.

## 3   Identifying Communication Channels in Low-Rank Subspaces

In this section, we test the hypothesis that model components like attention heads communicate through signals in low-rank subspaces of the residual stream and that we can find these signals in the weights themselves. We investigate one case of each type of composition outlined in Section 2 and find positive evidence for this hypothesis with query and value composition in 1 and 2 dimensional subspaces, but not key composition (see Appendix C). Because we are able to localize the query and value signals to such small representation spaces, we find that we are able to interpret and control these features with intervention experiments in Section 4.

### 3.1   Composition Score

The Composition Score (CS) introduced in Elhage et al. [2021] is a weight based metric of how much two weight matrices 'talk' to each other when they are separated by layers. That is, $\mathbf{W_1}$ might write information into a subspace in the residual stream that is read out by $\mathbf{W_2}$. In Query composition for example, $\mathbf{W_1}$ is the OV matrix of some head, and $\mathbf{W_2}$ is the QK matrix.

$$CS(\mathbf{W_1}, \mathbf{W_2}) = \frac{||\mathbf{W_1}\mathbf{W_2}||_F}{||\mathbf{W_1}||_F * ||\mathbf{W_2}||_F} \tag{1}$$

We take advantage of the fact that circuit analysis in works like Wang et al. [2022] tells us that, for example, head 3.0 (duplicate head) interacts with head 7.9 (inhibition head) through value composition and 7.9 with 9.9 (mover head) in query composition.

### 3.2   Composition with Decomposed Component Matrices

We initially use the composition score in Equation 1 to try predict these interactions in the weights, but find these results are largely noisy and uninterpretable. This is briefly demonstrated in Appendix I. We find that despite empirically knowing interactions exist between heads, we do not find they reliably have higher composition scores than any random head. Although the composition score has been shown to be useful on small toy models [Elhage et al., 2021], previous work has also shown that on larger models, the signal the composition score conveys is extremely noisy [Singh et al., 2024]. Therefore, we turn to the Singular Value Decomposition (SVD), defined as $\mathbf{W} = \mathbf{USV^T}$, on the QK and OV matrices to decompose the attention heads into orthogonal components which determine the input and output spaces of the matrix. This allows us to individually view subspaces

read from/written into ordered by the amount of variance of the transformation of the matrix they account for. This helps us answer our original hypothesis that model components communicate across layers in low-rank subspaces of the residual stream.

If $d$ is the dimension of the residual stream (768d in GPT2) and $h$ is the dimension of an attention head (64d in GPT2), $\mathbf{OV}$, $\mathbf{QK} \in \mathbb{R}^{dxd}$ and are both rank-$h$. This is because attention heads project down from the residual stream to $h$ (e.g., the job of the $\mathbf{V}$ matrix) and then back up to $d$ (e.g., the job of the $\mathbf{O}$ matrix). Therefore, there are only $h$ non-zero singular values for each matrix.

Equation 2 shows a useful identity of the SVD: we can rewrite some weight matrix $\mathbf{W}$ as the sum of the outer products of the left and right singular vectors, scaled by the corresponding singular value.

$$\mathbf{W} = \sum_{i=0}^{h} s_i * \mathbf{U_i} \otimes \mathbf{V_i} \tag{2}$$

Rewriting the original matrix in this way is useful because we can now use the sum of any subset of component matrices in the composition score (Equation 1). Let the zero-th component of head 3.0 be written as 3.0.0. We can write the composition score between the 3.0.0 OV matrix and the 7.9 OV matrix as $CS(\mathbf{OV}^{3.0.0}, \mathbf{OV}^{7.9})$. Since each component matrix is an outer product of two vectors, each matrix has rank-1. This gives us a way to disentangle the full signal of a head into the sum of its component rank-1 matrices, or the subspaces that the head is able to read from/write to.

We find that decomposing weight matrices this way is very effective at cleaning up the composition score signal. We find attention heads that have very high relative composition scores with one component matrix of another head. For example, the second component of head 8.6 (referred to as 8.6.2) composes far more with mover head 9.9 than any of the other 63 components in 8.6 (5 standard deviations higher than the average) or when considering the full matrix as in $CS(\mathbf{OV}^{8.6}, \mathbf{QK}^{9.9})$. The bottom graphs in Figure 2 show these results for the inhibition heads. All of the inhibition head exhibit a similar phenomenon of single component dominance. The duplicate token head 3.0 also value-composes with inhibition heads in a similar way, using two components (3.0.1 and 3.0.2) far more than any other. Results are shown in Appendix D.

We can also use this decomposition to find specific pairs of heads that talk more than others. With the knowledge that two heads talk through a specific component of one head, we can find the other heads that communicate through this pathway. Doing so lets us find the signal encoding almost the full IOI circuit in GPT2-Small directly from the weights, without running the model. We outline these results in Appendix I.

We interpret these as communication channels between heads, but we would still like to establish these channels as directly affecting downstream component's behavior. We verify this is the case through a weight editing in Section 3.3 and through activation interventions in Section 4.

### 3.3 Model Editing

In the previous section, we found that within a given head, individual component matrices encode a much stronger composition signal than that encoded by the global matrix. This makes the composition score a much more useful tool than when only considering full-rank matrices. In this section, we verify that these identified components are indeed communication channels that carry causally important signals for model behaviors. We first look at the inhibition head channels.

### 3.3.1 The IOI Dataset and Inhibition Score

Because the behavior of the inhibition heads was initially described on the Indirect Object Identification (IOI) dataset in Wang et al. [2022], we explore the inhibition communication channels on that domain first. An example of the dataset is as follows: *"Then, John[S1] and Mary[IO] went to the store. John[S2] gave a drink to"*. Here, the two name options are possible, but generating "John" does not make sense. The role of the inhibition heads are to tell the mover head (9.9) to attend less to the first John token (and as a result copy the remaining Mary token). We thus define the **Inhibition Score** as the degree to which the mover head prefers attending to the IO token (Mary) over the S1 token (the first John). This is simply the attention score to the IO minus the attention score to the S1. Intuitively, full attention to the IO token would give a score of 1.0, -1.0 would be full attention to S1 (inverse inhibition), and 0.0 would be equal attention to both (no inhibition).

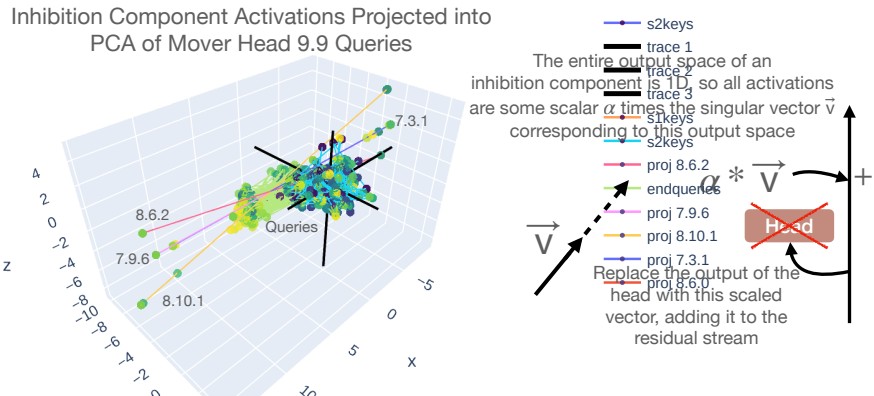

Figure 3: Because component matrices are rank-1, their output spaces are 1D and interpreting them becomes easier. On the left, inhibition component activations go to either side of the origin , and selectively inhiibt the name in either position one or position two in the IOI task. We can scale a vector lying on this line by some scalar alpha and observe how this changes behavior when we add it to the residual stream, or replace the output of an attention head with it (right), which we show in Figure 4.

### 3.3.2   Results

Our editing technique is simply to zero out one component at a time test how this affects copying behavior across a dataset of IOI examples. One way to think about this is zeroing out one singular value of e.g., the OV or matrix, or subtracting one of the component matrices from the sum in Equation 2. We must make the edit to the decomposition and then split the matrices back out so that we can run the model. Given $\mathbf{OV} = \mathbf{USV}^T$, after zeroing out some singular value in $\mathbf{S}$ (forming $\mathbf{S}'$) we can set the Output and Value matrices to be $\mathbf{U}\sqrt{\mathbf{S}'}$ and $\sqrt{\mathbf{S}'}\mathbf{V}^T$, respectively. In the top graphs of Figure 2, we show that removing the speculated communication channel from the inhibition heads almost always results in a significant decrease in the model's ability to pass the inhibition signal, with the exception that changing 7.3 does not have a strong effect on its own. In general removing the single component with the highest composition score reduces the Inhibition score by 7-14%, and it is important to consider that this is only when changing a single component in one head at a time. We perform additional experiments with removing/modifying multiple of these components in Section 4 and Appendix E. Thus, we have both behavioral and weight based evidence converging on the interpretation of these subspaces as meaningful communication channels.

## 4   Communication Channels Carry Interpretable Content-Independent Signals

We present further evidence that the communication channels we identify in the model weights carry causally important signals for affecting model behaviors, but also that they carry content-independent signals which are easily controllable and interpretable.

Because the component matrices are rank-1 (Equation 2), their outputs lie entirely on 1D subspaces. This subspace (in our implementation) is the right singular vector corresponding to the index of the component matrix multiplied by some scalar. Since we have shown that these communication channels have a significant impact on downstream performance, a natural question is how information (such as inhibition) is represented along such a simple feature.

### 4.1   Interventions on Communication Channels

In order to better understand the representations passed through communication channels, we design interventions that add to or replace information from certain heads with vectors that lie on the output space of communication channels. Figure 3 provides an outline of the approach. Since a single

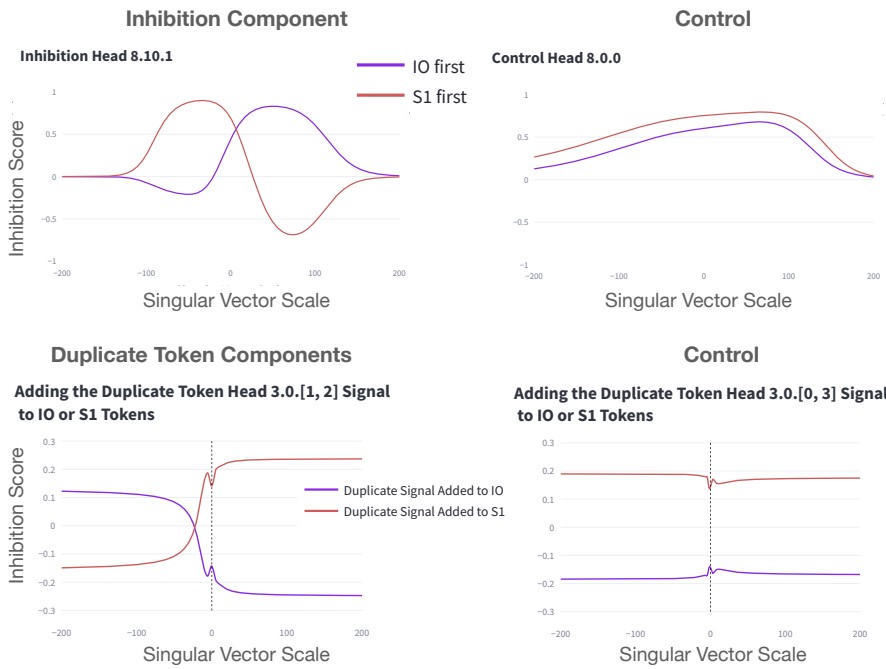

Figure 4: We find that the 1D inhibition components and 2D duplicate token components finely control which name is avoided by the mover head. On the top, we can selectively inhibit either the first or second name depending on how we scale a vector lying on the 8.10.1 output space. This is strictly controlling relative position. On the bottom, we find that adding or removing duplicate token information from the duplicate channel at the IO or S1 tokens also effectively modulates which name is inhibited. Neither random heads, nor non-communication channel components exhibit these same effects (right). See Appendix E for results on other heads.

component is rank-1, we can set the output of some head to be a point on the output space line and see how information changes along it.

Our dataset contains 200 examples from the IOI task. We have 100 examples where the IO token is the first name ("Mary and John...John gave a drink to") and 100 where the S1 token comes first ("John and Mary... John gave drink to").

On inhibition heads, we find that scaling a single component at a time is enough to switch the attention of the mover head to the other name. The inhibition score is highest when the S1 token is inhibited, but as Figure 4 shows, downscaling 8.10.1 only increases inhibition when S1 is first, and *decreases* it when IO is first. The opposite is true for upscaling the component. This tells us that the component is passing a positional signal: either inhibit name one or name two. This is consistent with what Wang et al. [2022] found, but our intervention shows that we can *completely* divorce the context from this ability. Since we are setting the head output to a scaled singular vector from the weights, we are bypassing the attention mechanism entirely, so non of the information on what to inhibit is coming from the value vectors of the IO or S tokens. This tells us that the model is capable of representing pointers, or storing bindings in the keys of earlier tokens that represent indexed lists, similar to the finding in Feng and Steinhardt [2023]. We explore this further in Section 5. The bottom of Figure 4 shows that scaling diagonally in a 2D subspace of the duplicate token head and adding the resulting vector to the residual stream of the IO or S1 token right before the inhibition heads also allows for modulating the selected name.

Although we get fine control over the attention of the mover head, we have not answered whether this has a real effect on the output behavior. Additionally, Makelov et al. [2024] argue that interventions on subspaces, such as the case here, are prone to a *subspace illusion* in which the effect is not what it seems. To address this, we measure the Fractional Logit Difference Decrease (FLDD) and Interchange Accuracy of patching the subspaces in the inhibition channel between minimally different IOI examples. GPT2-Small achieves an FLDD of 97.5% and and interchange accuracy of 35%.

Makelov et al. [2024] report a baseline that achieves -8% FLDD and 0.0% Interchange. By taking the gradient to directly optimize a single vector for this task they achieve 111.5% FLDD and 45.1% Interchange. Our results in comparison, support the view that these 1D subspaces are a primary mechanism controlling name selection. In Appendix G we also find that these inhibition signals are active broadly during next work prediction. On OpenWebText [Gokaslan and Cohen, 2019], we find that the inhibition heads are primarily active in lists and settings where repetitions should be avoided; for example, in comma separated lists (attending from commas to previously seen items). A natural followup given the IOI results and these observations is to explore their role as an indexing function, which we perform with the Laundry List task.

## 5 The Inhibition Channel is Crucial in Context Retrieval Tasks

Now that we have established the existence of communication channels and their causal role in model behavior, we can revisit our motivating example on prompt sensitivity in the Laundry List task. Figure 1 shows an example of the task and an example of an arbitrary seeming failure to minor variation in the order of presented items. In this section, we explore the hypothesis that the inhibition communication channel presented so far plays a critical role in how the model selects the right context token to generate next, and reveals a capacity limit that causes the model to fail as the number of objects increases.

### 5.1 Laundry List Task

We propose a synthetic task that is designed to activate the inhibition heads, but allows us to test their effect on an arbitrary number of candidate tokens. An example is given in Figure 1 and more details on how we generated the data are in Appendix F. Each example first mentions N objects, then N-1, and the model must predict the missing item. We create a dataset of 250 prompts for each value of N from 3 to 20.

To complete this prompt, the model needs to recognize the item missing from the second list and retrieve it from the first list. By shuffling the second list and using a different sentence format, simple mechanisms (like pattern matching with induction heads) can not be relied on solely to solve the task. With this setup, we can very naturally increase the number of objects being considered. This lets us test not only *if* the inhibition heads are used for indexing candidates, but whether this mechanism's behavior changes as it is strained by the number of comparisons that need to be made. The model has a strong bias to predict the last item, regardless of whether it is correct or not, which causes performance to drop, so we are also curious if the mechanism connects to that as well.

We find that the inhibition heads are highly active (large attention on non EOS tokens) on this task when predicting the last object, like the analysis in Section 4 would predict. The heads attend to and inhibit the occurrences of all of the repeated object tokens.

### 5.2 Intervention Experiments

We repeat the inhibition component intervention from Section 4 where we scale the components in the inhibition channel. In the multi-object (>2) case, we find that scaling only one component at a time does not give enough expressive power to change the mover head's attention to reach all of the objects (Figure 5, left). Instead it prefers to attend to either the first or last object, seeming to chunk the remaining objects together as a single point along the line. The bias to ignore information in the middle has also been observed by Liu et al. [2024].

We traverse the 3D space spanned by the top three inhibition head components (7.9.6, 8.6.2, and 8.6.10) and measure how this changes where the mover head attends, and what the model's final prediction is[4]. We leave out 7.3 because we found that it changes inhibition performance the least (Figure 2), and visualizing with only 3 dimensions is much simpler. In Figure 5 (middle), we visualize this traversal with one dot representing a point in the space that we query. We run the entire dataset with the inhibition components set at this point, and the color represents the index of the object that the mover head attends most to. We find that structure emerges as we add items, where areas of this

---

[4]We test in increments of 10 from [-100, 100] along each axis, including every combination. it's possible this is not the optimal range

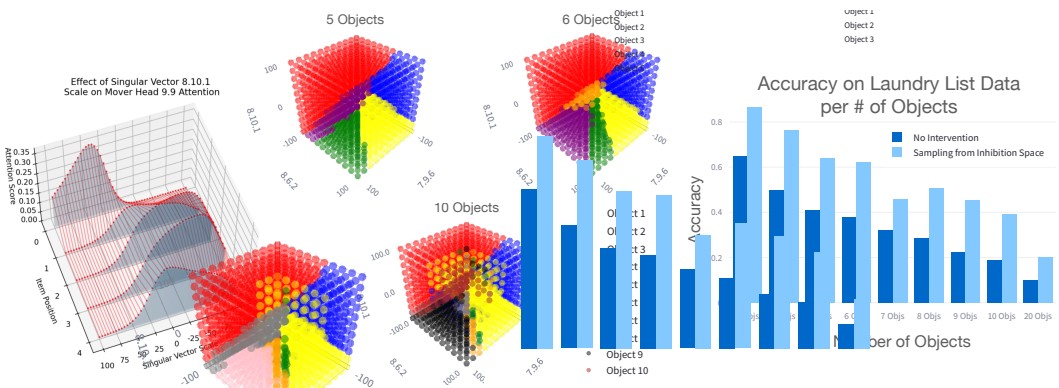

Figure 5: Scaling the inhibition component for a single head (here 8.10, left) is not expressive enough to get the mover head to index between the various objects. Scaling the top three inhibition components (middle) gives us enough expressive power to selectively attend to one of the objects. Here, one dot represents a run on the corresponding dataset and the color represents the index of the object the mover head pays the most attention to on average. A surprising structure emerges that partitions the space according to the index of the objects. However, the neat structure begins to break down as the number of objects grows around 10 or higher, and affects the mover head's ability to attend to the right object, which impacts accuracy. Right: Accuracy improvements as a result of sampling from inhibition space. The model becomes much more capable of handling a bigger number of objects in that the accuracy for N objects after the intervention is about as high as the unaltered model when it sees N/2 objects. However, the representational power of the inhibition channel reaches capacity as the number of objects increases, and performance can not improve as much.

space represent the first and last object, and wedges fill in the space for each item that gets added. Eventually (around 9 or 10 items) these wedges get small and start to fracture (Figure 5, middle bottom). We connect these two phenomena to the performance of the model: the bias to predict later objects, and the inability to handle longer lists. We believe the model is not capable of traversing this space well enough on its own, even though it learned to represent it, and longer lists cause worse performance because the space gets fragmented into smaller and smaller areas that repersent each item.

We design an intervention where we set the model components in a certain area of the 3D space, depending on the index of the correct answer and test how much this improves performance. In Figure 5 (right), we show this causes a sharp increase in the accuracy of the model: 3 object accuracy goes up from 64% to 86%, and 8 objects goes up to about 51%, about the level of 4 objects in the unmodified case. Therefore the inhibition channel we identified seems to form part of a more generic, content-independent subcircuit for indexing items in the previous context.

## 6  Related Work

There has been significant focus on disentangling features from the representations of language models and vision models [Olah et al., 2020]. Features have been analyzed at the neuron level [Gurnee et al., 2023, Mu and Andreas, 2020, Dai et al., 2022, Tang et al., 2024] Sparse Autoencoders and Dictionary Learning [Bricken et al., 2023, Mallat and Zhang, 1993, Cunningham et al., 2023] attempt to deconstruct activations into more primitive features [Rajamanoharan et al., 2024], which is similar in spirit to our decomposition. Park et al. [2023] propose a unification of several perspectives on the linearity of featuers. The Superposition Hypothesis Elhage et al. [2022] posits that linear features are encoded in interfering ways. Our method is similar in flavor of disentangling tangled features to make them easier to read off of the weights.

The SVD has been used for interpretability and weight based analysis in the past [Sule et al., 2023, Praggastis et al., 2022, beren and Black, 2022]. Martin et al. [2021], for example, use the SVD on weight matrices to measure weiht quality to predict generalization performance and whether or not a model is well-trained. The SVD has been used in the training of LoRA modules [Hu et al., 2022],

as well as in other finetuning methods [Bałazy et al., 2024, Feng et al., 2024, Wang et al., 2024, Sun et al., 2024, Guo et al., Karimi Mahabadi et al., 2021]. Our work analyzes model properties that may facilitate these methods to work. **?** is perhaps the first to report that pretrained models have a very low intrinsic dimensionality, which helps explain and support our claim that we see so many low rank communication channels in large models. We are therefore excited about the connections between this line of work and work on the linear representations in language models [Elhage et al., 2022, Park et al.], which argues that features in LMs are represented as directions in space. There has been recent work in studying interactions between components in finer-grained feature spaces in the past [Kim et al., 2018, Marks et al., 2024, Geiger et al., 2023, Lepori et al., 2023, Zhang et al., 2024], whereas our approach begins by first analyzing the weights to find substantial connections in subspaces without requiring any data. Future work could connect these two, to find circuits, subnetworks, and/or directions in space for certain behaviors informed by connections in the weights, which alleviates the concern of finding these things from scratch. We outline preliminary results for such an approach for novel circuit discovery in Appendix I.

## 7 Discussion & Conclusion

Due to the recent and rapid success of language models, there is growing interest in understanding how they are able to use language so flexibly and solve difficult tasks. Our results contribute positive evidence that intricate content-independent structure emerges as a result of self-supervised pretraining. Although similar types of structure were previously thought to be impossible or unlikely to emerge in LMs [Lake et al., 2017], there is emerging evidence that LM pretraining encourages models to organize mechanisms into neat subprocesses [Feng and Steinhardt, 2023]. We use weight decomposition to uncover such structure and contribute to a fundamental understanding on how models route information between layers, a core part of understanding feature representation in models. We also show that low-level mechanisms such as those studied here can make real predictions about prompt sensitivity, a problem that has long plagued the robustness of LMs. The method we employ for weight analysis also holds promise for inference time steering, model editing, and automatic circuit discovery. We hope our work promotes future research on the interpretability of neural networks as well as their responsible deployment, and practical capabilities.

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

## A   Limitations

A limitation of our approach is that we are relying heavily on previous knowledge of the language model that we are using (GPT2-Small), which has been extensively studied. However, the insights that we are able to glean by building on this foundation of knowledge we view as more reason to approach interpretability work as building directly on model-specific knowledge. Additionally, our findings may be able to feed back into automating interpretability of new models. Another limitation of our approach is the inability to calculate query and key composition scores with models that implement relative positional embeddings like RoPE [Su et al., 2023] because of the non-linearities between the Query and Key products preventing QK to be calculated cleanly. It may be possible to simply take the composition between the Q and K matrices individually, but we do not experiment with that extension here.

## B   Larger Models' Performance When Increasing the # of Objects in the Laundry List Task

In the main text, we consider models that are quite small by today's standards, and it is reasonable to wonder whether the inability to handle more objects in context goes away with model scale. In Figure 6, we show that even up to 6.9B parameters, OPT [Zhang et al., 2022] and Pythia models also struggle with increasingly many objects, although degrade more slowly with scale. Regardless, we are *not* claiming that we have identified a problem that can not be solved through scaling, nor is that of particular interest in this work. Increased capacity is likely to yield mechanisms with higher capacity to handle more objects. What is of greater interest for future work would be to identify whether larger models use fundamentally different mechanisms than the one we identified here to solve the task.

## C   The Composition Score with and without Weight Decomposition

We include some examples showing outlier components in value and query composition but not with induction head key composition in Figure 7.

## D   Duplicate Token Heads

We focus on the inhibition communication channel in the main paper and do not show the where the duplicate token channel comes from. In Figure 8, we show that two component matrices in duplicate token head 3.0 (3.0.1 and 3.0.2) compose strongly with inhibition heads (7.9.6 shown here). On long sequences of random tokens, we show that these direction encode whether or not a token has been duplicated.

## E   More IOI Interventions

Additional interventions testing the efficacy of inhibition heads to change behavior of a downstream mover head are provided in Figures 9 and 10. We include the remaining inhibition heads and additional control heads for the component scaling experiment from Section 4 in Figure 11

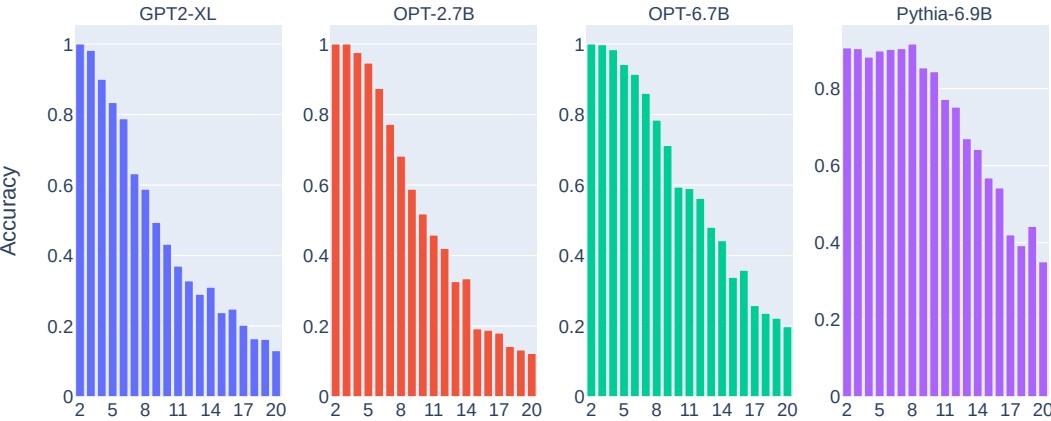

Figure 6: Larger models also degrade performance rapidly as we increase the number of objects in the Laundry List task, although more slowly than smaller models. Pythia 6.9B retains strong performance up to around 10 objects.

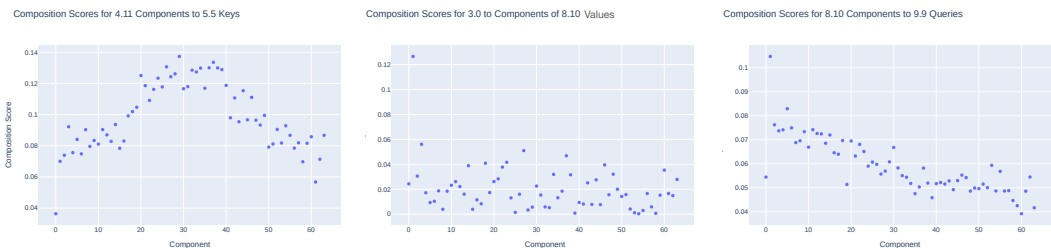

Figure 7: Examples of composition scores of individual components with other heads. 4.11 is a previous token head, 5.5 is an induction head, 3.0 is a duplicate token head, 8.10 is an inhibition head, and 9.9 is a mover head. We find that there are large outlier components in value and query composition, but not in the induction head, thus motivating our focus on those heads in the main text.

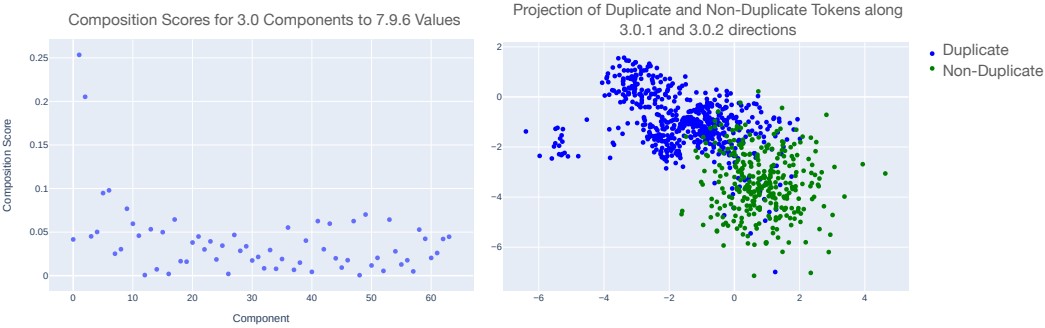

Figure 8: Left: Composition scores between each component of duplicate token head 3.0 and inhibition component 7.9.6. Components 1 and 2 are clearly outliers. Right: on long contexts of random tokens with inserted duplicates, we find that these directions separate duplicates from non-duplicates quite well. This leads us to believe that these two components form a duplicate communication channel. Our results in Section 4 support this interpretation.

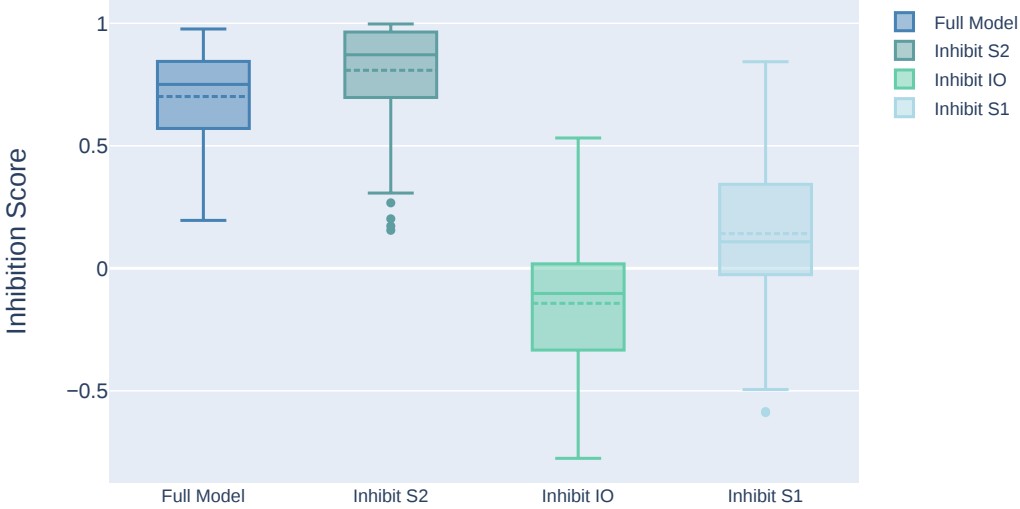

Figure 9: Effect of applying the top component interventions at the same time to some token. We can control the inhibition by selecting which token these components attend to. Higher score means more inhibition on S2, and lower score means more inhibition on IO.

## F Laundry List Data Generation

The Laundry List task is a leave-one-out task where the model must identify the object that was not mentioned. Each input is two sentences (see Figure 1 for an example). The first sentence lists objects that need to be purchased and the second describes the order that they are to be bought in, with the next token prediction being the item from the first list that is to be bought last. This setup allows us to freely shuffle the order of the information provided to the model as well as vary the number of objects presented in each example. There are 22 objects that can be sampled, given below: ''pencil", ''notebook", ''pen", ''cup", ''plate", ''jug", ''mug", ''puzzle", ''textbook", ''leash", ''necklace", ''bracelet", ''bottle", ''ball", ''envelope", ''lighter", ''bowl", ''apple", ''pear", ''banana", ''orange", ''steak" .

The first sentence can start a few ways, chosen randomly: ' Today,', ' Tonight,', ' Tomorrow,', ''. And the second sentence start can be chosen randomly: ' First,', '', ' When I go,', ' I think'

## G Inhibition Heads Behavior on Open-domain Text

We find that inhibition heads are consistently active on tokens about to predict the continuation of a sequence ("serotonin, dopamine, and...") and attend to previous items in that sequence, consistent with their role in IOI and Laundry List. We therefore argue that the role the inhibition mechanism plays in both IOI and the Laundry List task performs this same operation in generic language modeling

## H Inhibition Mechanism in Pythia and Training Progression of Inhibition

We verify both that communication channels appear in other models, and that inhibition is a more general mechanism that that just appearing in GPT2. To show this we analyze Pythia-160m [Biderman

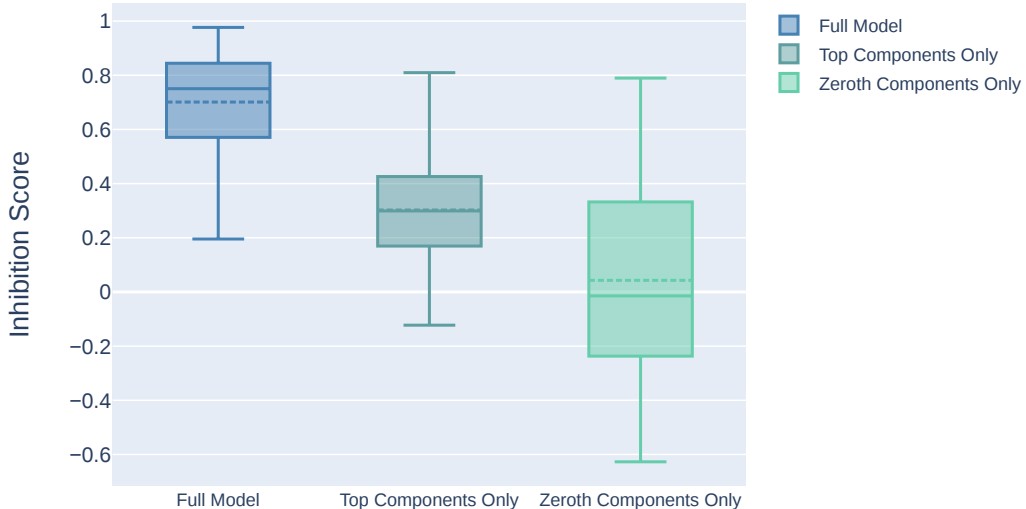

Figure 10: Effect on the inhibition score when removing components from the inhibition heads. If we only take the top-1 composing heads that affect the inhibition score (circled in Figure 2) we retain close to have of the average inhibition score (0.7 to 0.3). If we only use the component matrices that correspond to the 0th singular value of the inhibition heads, which represents the subspace most strongly written to by the head, the average inhibition score is only 0.04. Recall that a negative inhibition score means placing more attention on the subject rather than IO token.

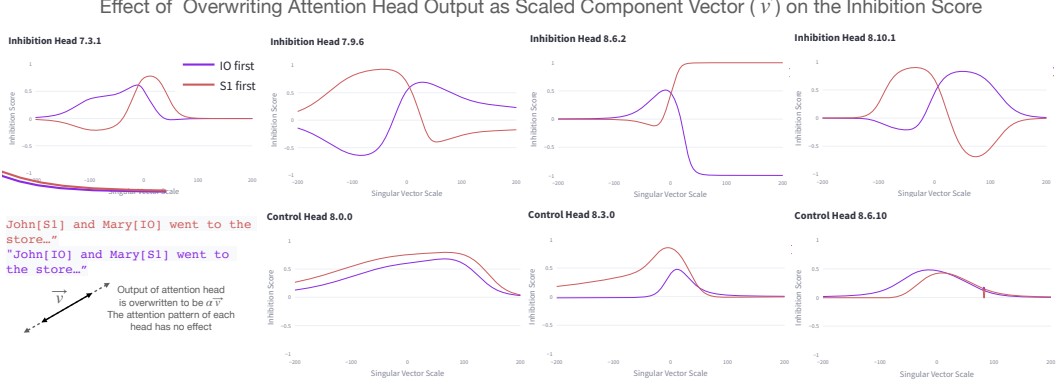

Figure 11: We intervene on the forward pass of the model by replacing the output of some attention head as the vector obtained by scaling a component vector by some scalar $\alpha$. By doing so, the actual attention head pattern has no effect on the downstream performance. We show the inhibition component vectors have the unique effect of controlling the position of the name being attended to by the downstream mover head (9.9). Random control and head components with other functions (like 8.3.0) do not have this effect.

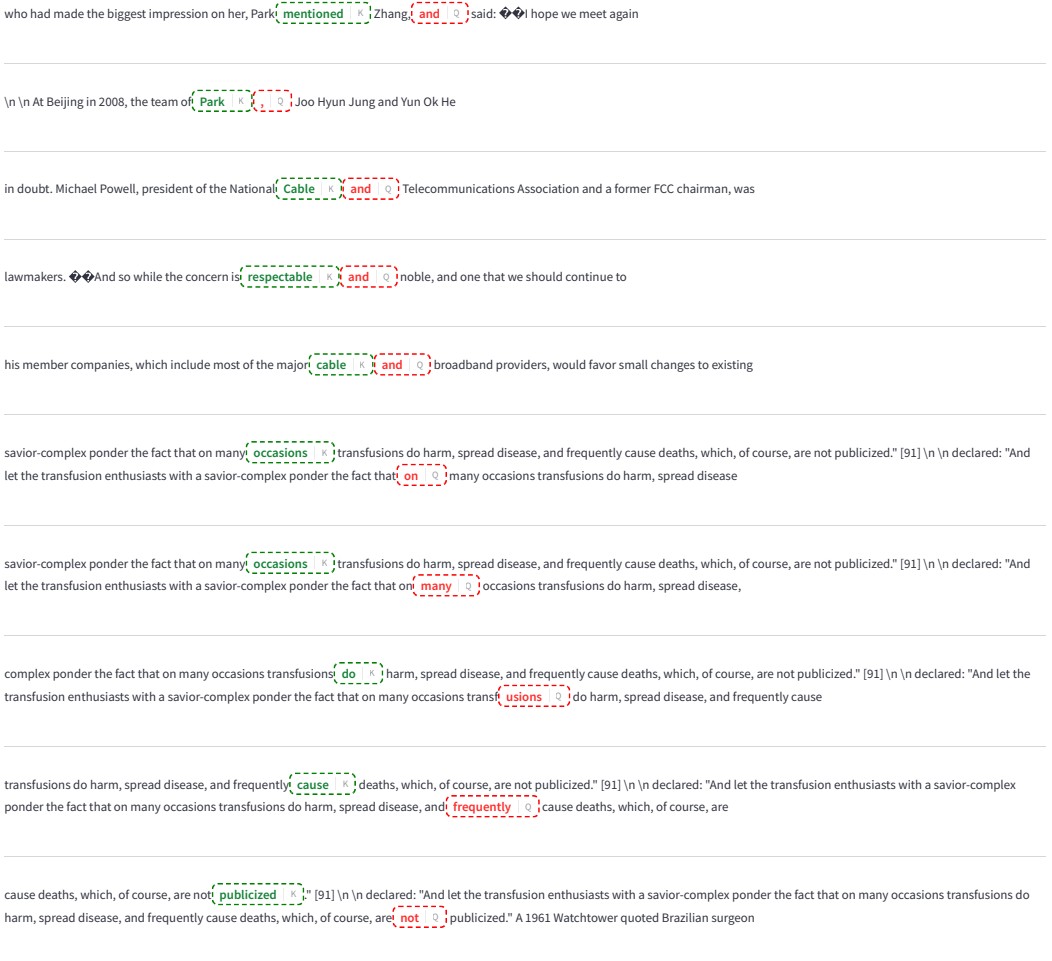

Figure 12: Examples of high attention in examples in OpenWebText-10K for head 8.6 in GPT-2

et al., 2023]. Because Pythia provides training checkpoints, we are also able to analyze the formation of the inhibition component we find to some extent.

## H.1 Path Patching on IOI

We perform path patching [Wang et al., 2022, Goldowsky-Dill et al., 2023] on Pythia-160m on the IOI task to see if the model also implements mover and inhibition heads. We find evidence for one inhibition head (6.6) in the model talking to a mover head (9.5). We find that like GPT2, the inhibition head communicates primarily through a single component, as is shown in Figure 16.

Additionally, an induction head (4.11) strongly value composes with the inhibition component 6.6.2 as shown in Figure 17.

## H.2 Training Progression of Inhibition Components

Because Pythia releases 144 intermediate checkpoints (per 1000 steps), we can track the emergence of the inhibition head during training. We saw that the inhibition component vector clusters Name1 on one side and Name2 on the other side, representing which name is being inhibited. Everything else ends up around the origin. Since we have minimal pairs of examples that differ only in the position of the name that should be inhibited, we can measure the Separability of the inhibition component

**GPT2 8.10**

need to recreate an exciting Turf War in your `room` [K]. You might want to keep them spread out in `your` [Q] collection, though, because when Inklings are

want to keep them spread out in your collection, `though` [K] `,` [Q] because when Inklings are together in one place

\n \n Inkling Girl \n \n Ink `ling` [K] `ling` [Q] girls hail from the city of Inkopolis

21, 2013 | Hammer Museum Courtyard \n \n `Free` [K] `and` [Q]

t have enough frontal cortex, you have an `excessive` [K] `,` [Q] discombobulated, inefficient, poorly wired

had then stolen the card numbers of the 40 million `customers` [K] and personal details of about 70 `million` [Q] customers. This damaged the company��s

May 2016. A port for the Nintendo Switch was `released` [K] in January 2019 and `was` [Q] ported by British studio Red Phantom Games,

their 'cut of the action'. \n \n Put `simply` [K] `simply` [Q] that's a huge brake on the US

the 2017 Hollywood Diversity Report), he landed the coveted `Jurassic` [K] job on the strength of a single film, `the` [Q] quirky 2012 Sundance sci-fi dramedy

the Summerhill LCBO. ��It was `delicious` [K] `and` [Q] sold really well, Tomney says

Figure 13: Examples of high attention in examples in OpenWebText-10K for head 8.10 in GPT-2

vector by making sure that if one name is in one cluster, the minimal pair example is in the other cluster. This is a measure of how well the inhibition head is structuring according to this idealized separation of the two names.

The component that we use is the 6.6.2 matrix from the fully trained inhibition head. We test parity by projecting the model's activations onto this matrix.

In addition we can measure how well the fully trained component matrix activates (or removes) the inhibition signal by adding or subtracting it from earlier checkpoints. These results are in

# I   Circuit Discovery with Static Weight Analysis

One of the core claims of this work is that we can find meaningful connections between attention heads by reading them off of the weights. In this section we extend this idea beyond the IOI circuit and use our method to show that we can find novel meaningful connections without running the model. First, we show that we can rediscover the IOI circuit directly from the weights using the decomposed composition score (without knowing anything about its function). Then, we show that we can identify new connections to the 9.9 mover head that is part of a different circuit. We finally run the model on open-domain data and perform causal experiments to begin to identify the functionality of this new circuit, which we implicate in retrieving relevant information from context. While the experiments we perform to identify functionality are not a replacement for a full circuit analysis, we at least show how our method can be used to pinpoint circuits in a network for further analysis. Future work can extend the analysis of the context-retrieval circuit we find here.

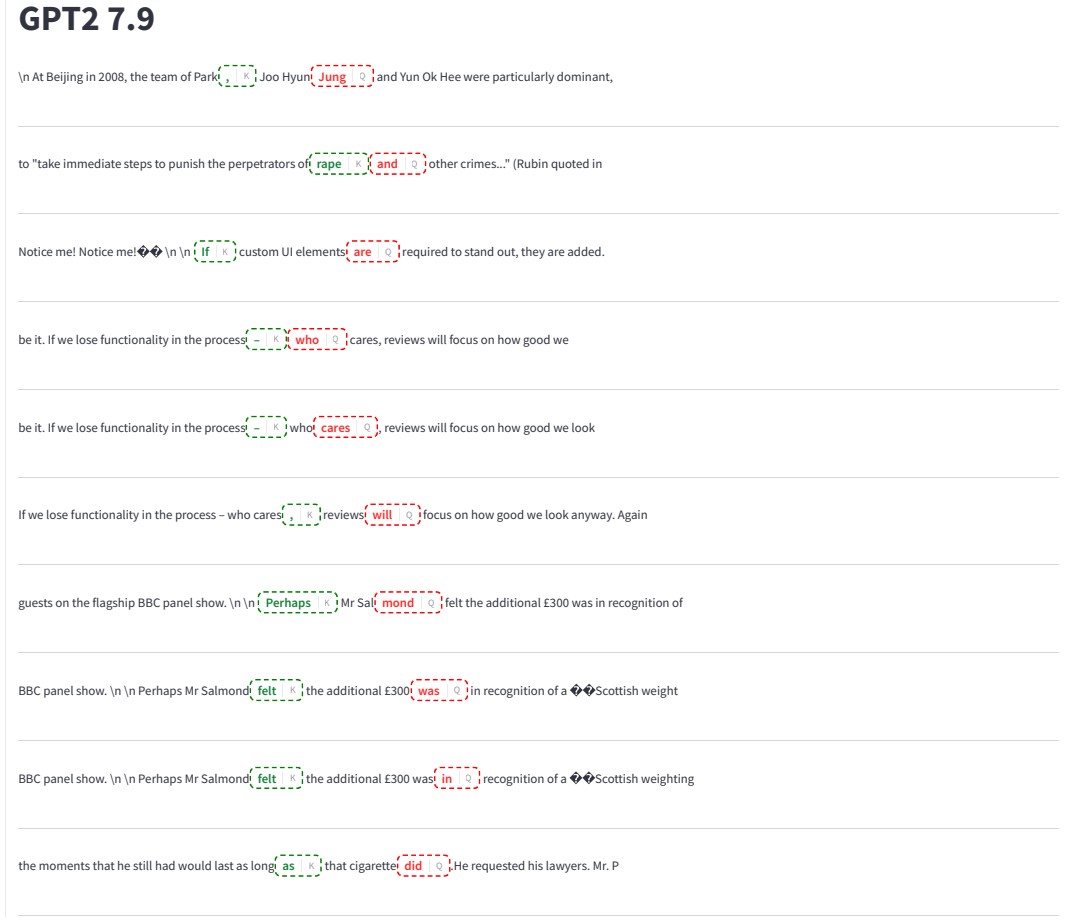

**GPT2 7.9**

\n At Beijing in 2008, the team of Park [ , | K ] Joo Hyun [ Jung | Q ] and Yun Ok Hee were particularly dominant,

to "take immediate steps to punish the perpetrators of [ rape | K ] [ and | Q ] other crimes..." (Rubin quoted in

Notice me! Notice me!◆◆ \n \n [ If | K ] custom UI elements [ are | Q ] required to stand out, they are added.

be it. If we lose functionality in the process [ – | K ] [ who | Q ] cares, reviews will focus on how good we

be it. If we lose functionality in the process [ – | K ] who [ cares | Q ], reviews will focus on how good we look

If we lose functionality in the process – who cares [ , | K ] reviews [ will | Q ] focus on how good we look anyway. Again

guests on the flagship BBC panel show. \n \n [ Perhaps | K ] Mr Sal [ mond | Q ] felt the additional £300 was in recognition of

BBC panel show. \n \n Perhaps Mr Salmond [ felt | K ] the additional £300 [ was | Q ] in recognition of a ◆◆Scottish weight

BBC panel show. \n \n Perhaps Mr Salmond [ felt | K ] the additional £300 was [ in | Q ] recognition of a ◆◆Scottish weighting

the moments that he still had would last as long [ as | K ] that cigarette [ did | Q ] He requested his lawyers. Mr. P

Figure 14: Examples of high attention in examples in OpenWebText-10K for head 7.9 in GPT-2

## I.1 Finding the IOI Circuit in GPT-2 Weights

To show the effectiveness of the decomposition at finding heads that communicate to a significant degree, we use the composition score only to find a large chunk of the IOI circuit from Wang et al. [2022] encoded directly in the weights. Figure 19 shows these results. The composition score after decomposing the inhibition head 7.9 more clearly reveals the communication between the in-circuit heads (circled in red) than if the composition score is used without decomposition. It is possible this approach can be built upon to find circuits in models without requiring the model to be run. We leave this for future work.

## I.2 Discovering New Connections

We now take this further to show that we can use the composition score to facilitate the discovery of new circuits directly from the weights. Although we can not yet identify function directly from the weights, we can identify components that communicate strongly (indicated by their composition scores) to know where to look for circuits that may explain other behaviors in the model.

We focus on one specific extension to the inhibition-mover subcircuit by looking for new connections into the 9.9 mover head QK circuit. We search for connections into this head because it seems likely that other heads besides inhibition heads control what the mover head attends to, but it has not been investigated in prior work. We look for component matrices that have a high composition score with the 9.9 QK matrix, and plot a heatmap of the highest composing component matrix per head in Figure 23. We find that there are several heads that have components that highly compose with 9.9 that have not been previously identified as affecting copying behavior. These are heads 6.1.1, 6.2.5, 6.7.3, and 8.3.0. We show the per component composition scores for these heads in Figure 24 and find

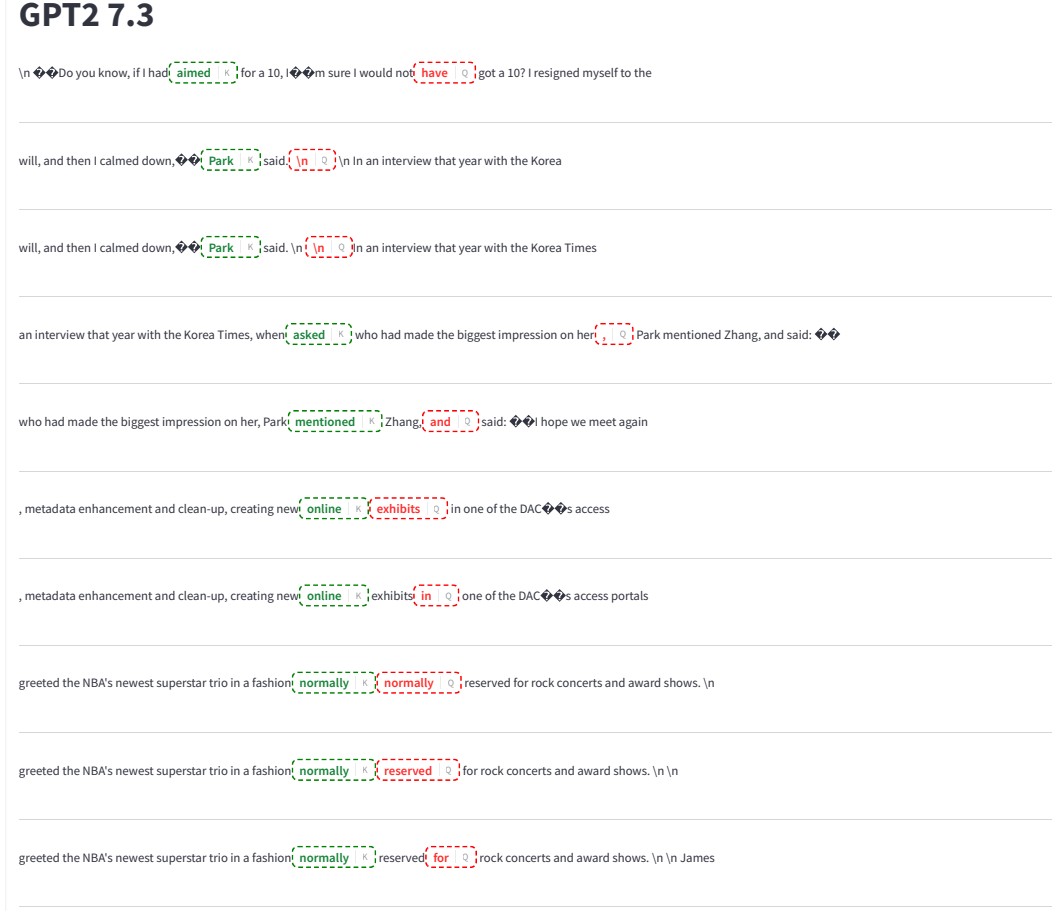

Figure 15: Examples of high attention in examples in OpenWebText-10K for head 7.3 in GPT-2

that they also contain individual components with very high relative composition compared to other components (with 6.2 components being distributed slightly more smoothly than the others).

### I.2.1  Attention Pattern Analysis of 6.1, 6.2, 6.7, and 8.3

We perform a similar attention pattern analysis as shown in Section G on these heads. We run the model on documents from OpenWebText-10K and look at contexts in which these heads attend strongly ($\geq .5$ of probability mass) to some token. We find that 6.2 is not interpretable in this way, as its attention patterns are more diffuse, but examples for the other heads are shown in Figures 21, 22, and 20. We find qualitatively similar patterns in these heads to inhibition heads, though with less selection for attending to previous items in lists. Head 8.3 is of particular interest because it has the strongest composition with 9.9, and we will focus on it for the remainder of this section. This head also has some particularly salient motifs in its attention patterns: much of the time it appears to attend to a token (or the last token in a multitoken phrase) in context that could be a plausible continuation of the current text. In the following section we explore what the possible function of this head is, and its connection to the 9.9 mover head.

### I.2.2  Attention Head 8.3 as a Relevant Context Head

The attention patterns in Figure 20 suggest this head plays some role in attending to relevant continuations of the current context. The strong composition to the 9.9 mover head QK circuit (Figure 23) also suggests it is modulating what the copying head attends to. But what is the actual function of this head? First, we establish that this head is not a mover or inhibition head, then provide evidence that it is a head involved in signaling relevant context that can be copied to the next token by the

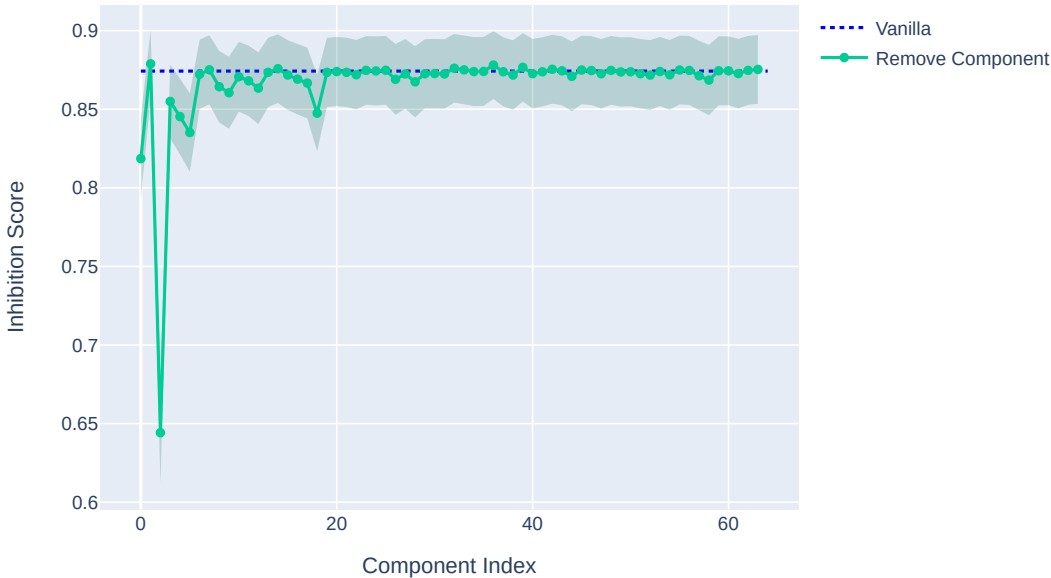

Figure 16: Pythia-160m also has a single component (6.6.2) in an inhibition head that dominates the inhibition signal.

mover head (9.9). We begin to outline how exactly the 8.3.0 component matrix interacts with the attention pattern of 9.9 with some success, but can not make strong conclusions because we can not control for every variable in the data we have available. We hope this work encourages future study on this head's interactions with mover heads as well as future work in static weight analysis more generally.

**8.3 is not a mover head**     It's possible that 8.3 acts like a mover head, copying the tokens it attends to into the residual stream, which incidentally causes 9.9 to do the same. We perform the IOI copying head test from Wang et al. [2022] and find that this is likely not the case. In this test, the output of an attention head is decoded into the vocab space. If the top token from this decoding is the same as the token it attends to, then it is "copying" that token. We find that 8.3 gets 0% on this test, while 9.9 gets 100%. Other inhibition heads also get 0%, so next we explore whether 8.3 could be interpreted as a type of inhibition head.

**8.3 is not an inhibition head**     We include 8.3.0 (the zeroth component in 8.3, which strongly composes with 9.9) in our analysis in Figure 11, and find it does not act like an inhibition head on the IOI task (although, as we could expect from the composition score, it does modulate the inhibition scores). Additionally, removing the zeroth component reduces the inhibition score on IOI by less than .01 (same experiment as performed in Figure 2). From these experiments, we can not rule out that 8.3 **never** acts like an inhibition, i.e., it could do so on a distribution of text we do not test, but evidence seems to point against this. Regardless, we can provide evidence for this head being involved with identifying relevant continuations (rather than preventing repetitions) in the next sections

**8.3 attends to *relevant* continuations of the current context**     Based on the observational evidence from the attention patterns on excerpts from OpenWebText-10K, we hypothesize that 8.3 will selectively attend to tokens that are logical continuations of the current text. To test this, we create a small synthetic counterfactual dataset of verbs and nouns to test 8.3's attentional preferences. Our intuition is that: given some mention of a noun $n$ in context ("pork"), and a verb $v$ later in context ("eat"), 8.3 will only attend to $n$ if it is an appropriate direct object of $v$. Therefore, in the contexts "I see the pork. I am eating" and "...I am drinking", 8.3 will attend from eating to "pork" strongly, but not from "drinking" to "pork", since this is not a typical object of the verb "drink". We generate 84

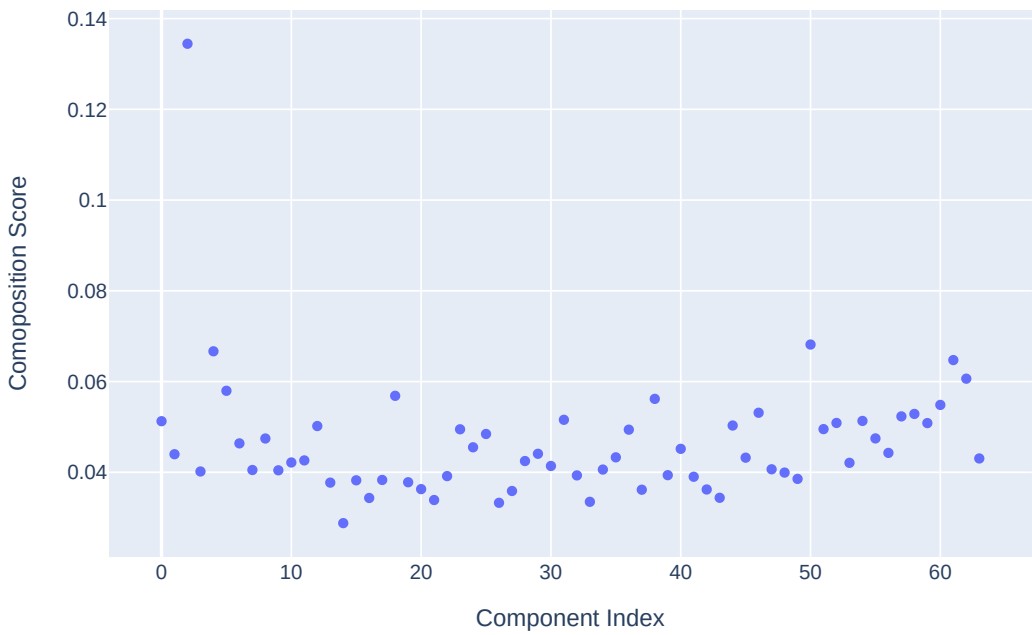

Figure 17: An example of an induction head (4.11) value composing strongly with the single inhibition component 6.6.2 in Pythia-160m, suggesting a circuit for controlling attention through mover head 9.5. We leave analysis of this for future work.

| Verbs | Likely Objects |
|---|---|
| cook | [carrot, cabbage, steak, pork, chicken] |
| eat | [carrot, cabbage, steak, pork, chicken] |
| drink | [water, juice, milk, soda, coffee] |
| read | [book, newspaper, magazine, comic, blog] |
| watch | [movie, TV, video, show, cartoon] |
| open | [door, window, box, jar, can] |
| write | [letter, note, script, report] |
| play | [game, piano, guitar, violin, drum, song] |
| paint | [wall, fence, door, floor, ceiling] |

Table 1: Verbs and appropriate objects used to test 8.3's attention patterns to relevant objects

counterfactual pairs of a verb and a likely/unlikely object using the verbs and nouns in Table 1. An unlikely object is any noun that does not appear in the verb's list of likely objects. We find that 8.3 attends much more strongly to nouns that are likely to follow the given verb. Results are shown in Figure 25.

**Causal Interventions on 8.3.0**    Next, we use the output space of the 8.3.0 component as a steering vector (since it outputs onto a line) to change the attention of 9.9, as predicted by the composition score to 9.9's QK circuit. We use excerpts from documents in OpenWebText-10K that highly activate 8.3 (Figure 20). We hypothesize that 8.3 plays some role in telling 9.9 to attend and copy to relevant context through its zeroth component matrix. Specifically, we test whether adding some constant times the 8.3.0 output vector to the residual stream will cause 9.9 to attend more or less to the token

**Training Progression of Inhibition Component Subspace and Mover Head Inhibition Score**

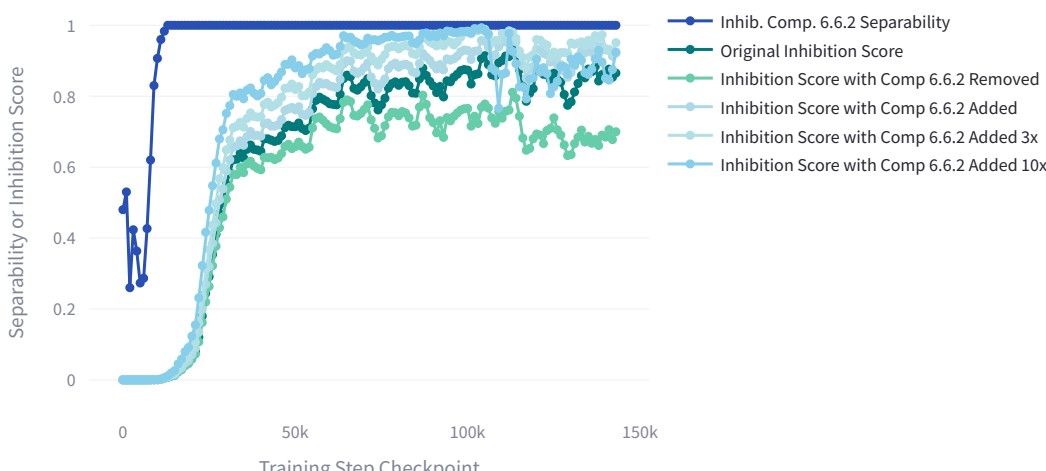

Figure 18: Pythia training progression of inhibition component (6.6.2) and effect of model editing. Adding the component matrix to the inhbition head strengthens the inhibition channel and improves the ability to use inhibition in earlier checkpoints, subtracting it makes inhibition weaker. Separability is simply the extent to which activations for IOI minimal pairs are split into clusters based on the order of names (IO, S1 or S1, IO).

8.3 attends to (following a similar methodology as in Figure 4). For instance, given the input "would be better to negotiate with David, to give [him] some money and to [allow]", 8.3 attends from "allow" to "him". We find that this does tend to be the case, although it is difficult to quantitatively measure. Our results in Figure 26 show per-example attention scores after adding 8.3.0 multiplied by -1000 to the residual stream. We see a slight increase on average, but the results are inconsistent across examples. One reason is because of multitoken phrases; e.g., for phrases like "the corridor", 8.3 will attend consistently to the last token in the phrase, here "corridor". But if 9.9 is to copy this phrase, it should attend to "the", not corridor. Our test does not cover this, and it isn't clear how to fairly handle each edge case. We suggest that some interaction like the one we've described here is taking place, but leave more rigorous analysis on the exact interactions between 8.3 and 9.9 for further study.

## J  More Information on Composition

### J.1  Value Composition: Duplicate Token Heads

Value composition dictates that the value vectors of an earlier head write information that affect the values of later heads. Duplicate token heads are a well established type of attention head that specialize in attending to duplicates in the previously seen context. That is, given the text "A B A", the duplicate token head will attend from the second "A" token to the first. The IOI circuit finds value composition between heads 3.0, a duplicate token head, and 7.9, an inhibition head.

### J.2  Query Composition: Inhibition Heads

Value vectors of earlier heads affect the query vectors of later heads, thus changing what they attend to. A canonical example originating in the IOI paper is with *inhibition heads*. These are a key part in a token copying circuit in which the value vectors of such heads prevent later query vectors from attending to the duplicated name in the IOI task. Mover heads (such as 9.9) We study these heads in greater detail in Section 5. Whatever a mover head attends to will be promoted as the next token prediction. An inhibition head tells a mover head to avoid attending to certain tokens, which is helpful when there are multiple options to generate.

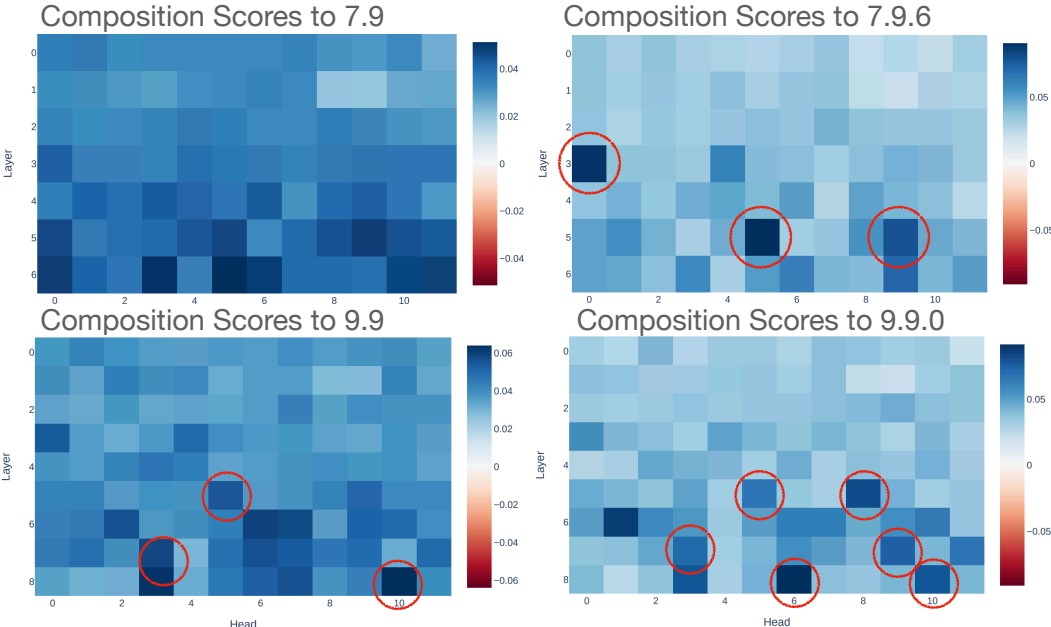

Figure 19: Decomposing weight matrices cleans up the composition score enough that we can start to read off components that belong to the IOI circuit without running the model. By starting with a known inhibition head component (7.9.6) we can find the heads that compose into that component and the heads for which the inhibition component composes into that belong to the IOI circuit from Wang et al. [2022]. Left graphs show the composition score without any decomposition, which is noisy. On the right, we find in-circuit heads (circled) qualitatively to stand out more. See Wang et al. [2022] for more details.

### J.3 Key Composition: Induction Heads

An induction head is a pattern completing attention head. For example, seeing the pattern "A B A B A" will cause the model to attend from the last A to the last B, since a pattern is present where B must follow A. The mechanism that typically implements induction heads requires a previous token head (which will always attend from the current token to the one right before it) to affect the key of a later induction head. At a later timestep the query of the induction head will notice the signal left in the earlier key, and choose to attend to it. We consider the key composition between the previous token head 4.11 to induction head 5.5 from the IOI circuit.

## K   Extra Laundry List Interventions

The results for scaling individual components across Laundry List datasets (varying number of objects) are in Figures 28, 29, 30, 31, 32, 33, 34, 35, and 36

We also include results from traversing the 3D inhibition subspace used in Figure 5 for a greater range of settings for the number of objects in Laundry List dataset examples. We test datasets set to have 3-10 objects and one dataset set to have 20 objects. The results are shown in Figure 27.

## GPT2 8.3

\n \n Henry K. Lee is a San Francisco Chronicle staff writer. E-mail: hlee @ sfchronicle.com Twitter: @hen

High School in Oakland. \n \n Henry K. Lee is a San Francisco Chronicle staff writer. E-mail: hlee@sfchronicle.com Twitter: @ henrykleeFreshen up your

hen up your amiibo collection with this Spl atoon amiibo series 3-pack! With Inkling Girl, Boy and Squid in fresh alternate colors, these amiibos are all you need to recreate an exciting Turf War in your room. You might want to keep them

purchasing or using free-standing ��seclusion cells �� or ��isolation booths, as they are called. With the new law taking effect, some schools are scrambling for a way around the provisions, so they can continue to place students in solitary confinement at their discretion. \n \n In

would be better to negotiate with David, to give him some money and to allow him to get on with his life. As

\n Termine und Deadlines \n \n 30 September 201 5 (23:59 UTC): Einreichungsende \n \n 16 . November 201 5 : Mitteilung �

Abbott, former Prime Minister of Australia, made a speech today at the Global Warming Policy Foundation in London. Here is a link to the speech. With this speech

Nobody can bring out, in rally after after rally, Americans who lost loved ones because these

rally, Americans who lost loved ones because these immigration laws were not enforced … or Americans who lost jobs because of unfair labor displacement," and not follow through on

of people bringing their own smartphones and tablets to the office (And, going back even further, the PC itself was often brought into companies on a departmental level.) A similar

Figure 20: Examples of high attention in examples in OpenWebText-10K for head 8.3 in GPT-2

## GPT2 6.1

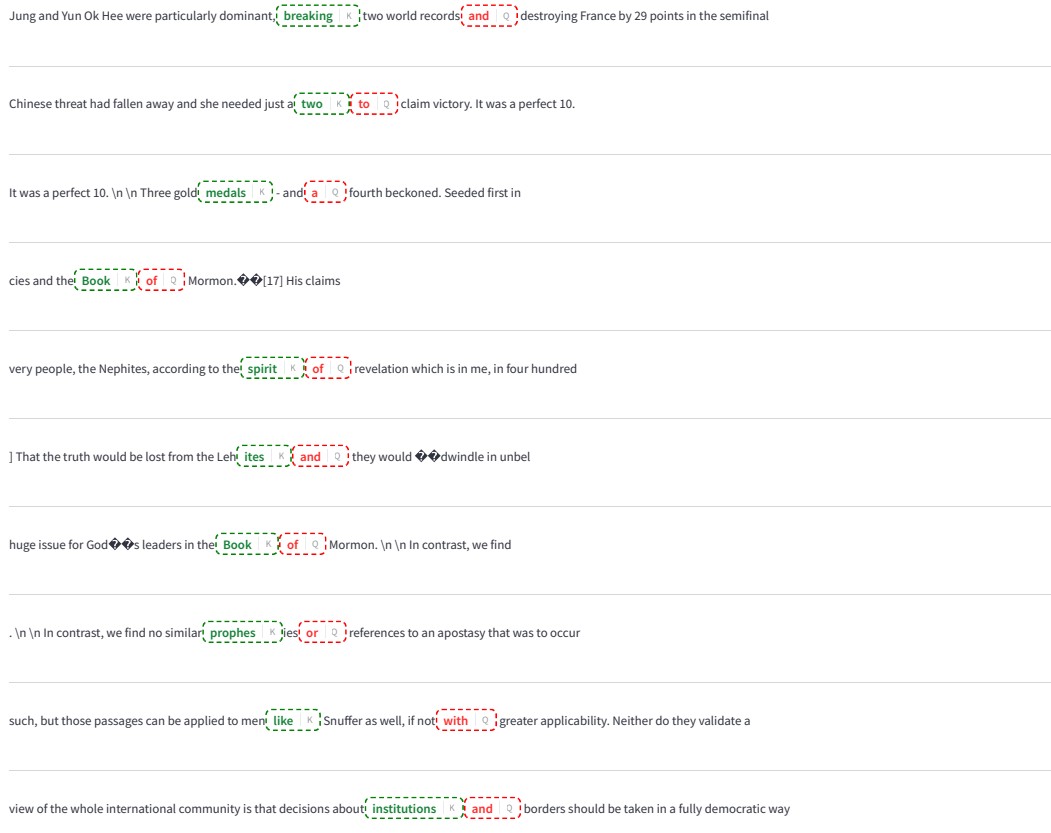

Jung and Yun Ok Hee were particularly dominant, breaking two world records and destroying France by 29 points in the semifinal

Chinese threat had fallen away and she needed just a two to claim victory. It was a perfect 10.

It was a perfect 10. \n \n Three gold medals - and a fourth beckoned. Seeded first in

cies and the Book of Mormon.��[17] His claims

very people, the Nephites, according to the spirit of revelation which is in me, in four hundred

] That the truth would be lost from the Leh ites and they would ��dwindle in unbel

huge issue for God��s leaders in the Book of Mormon. \n \n In contrast, we find

. \n \n In contrast, we find no similar prophes ies or references to an apostasy that was to occur

such, but those passages can be applied to men like Snuffer as well, if not with greater applicability. Neither do they validate a

view of the whole international community is that decisions about institutions and borders should be taken in a fully democratic way

Figure 21: Examples of high attention in examples in OpenWebText-10K for head 6.1 in GPT-2

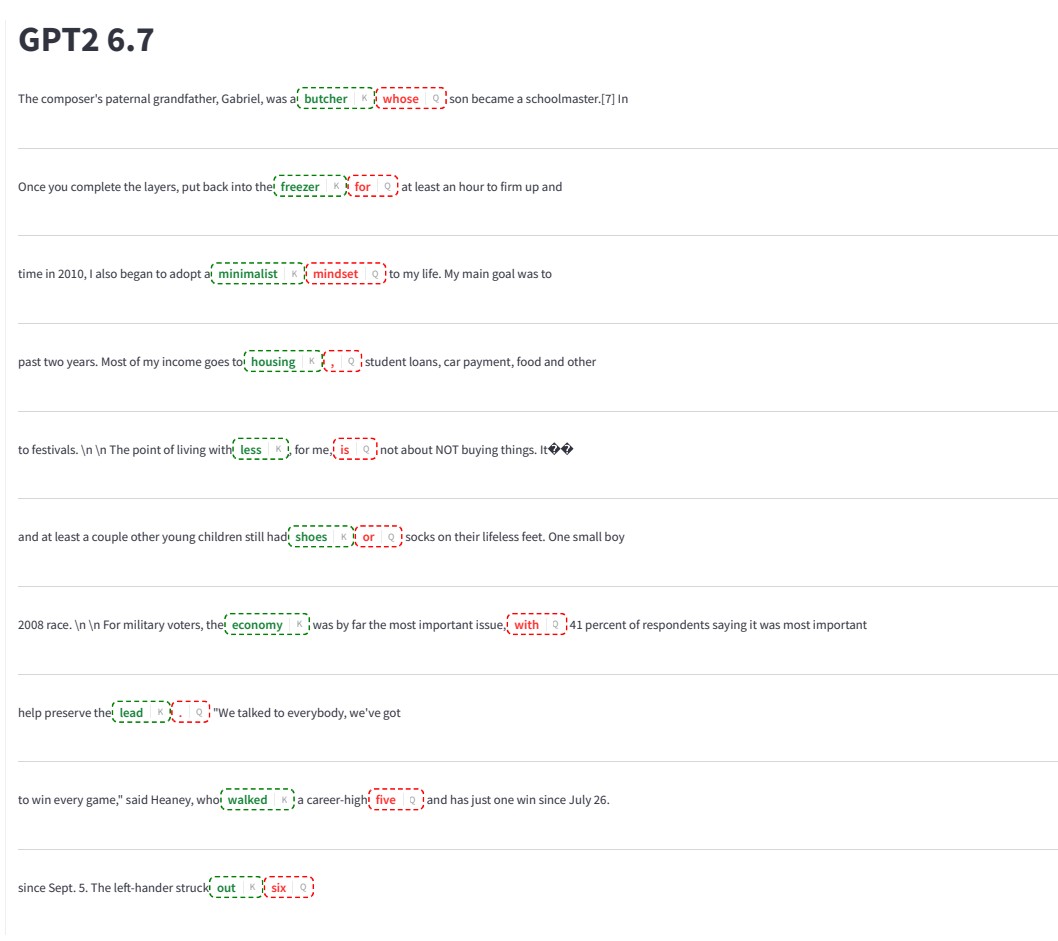

## GPT2 6.7

The composer's paternal grandfather, Gabriel, was a [butcher] [K] [whose] [Q] son became a schoolmaster.[7] In

Once you complete the layers, put back into the [freezer] [K] [for] [Q] at least an hour to firm up and

time in 2010, I also began to adopt a [minimalist] [K] [mindset] [Q] to my life. My main goal was to

past two years. Most of my income goes to [housing] [K] [,] [Q] student loans, car payment, food and other

to festivals. \n \n The point of living with [less] [K] for me, [is] [Q] not about NOT buying things. It

and at least a couple other young children still had [shoes] [K] [or] [Q] socks on their lifeless feet. One small boy

2008 race. \n \n For military voters, the [economy] [K] was by far the most important issue, [with] [Q] 41 percent of respondents saying it was most important

help preserve the [lead] [K] [.] [Q] "We talked to everybody, we've got

to win every game," said Heaney, who [walked] [K] a career-high [five] [Q] and has just one win since July 26.

since Sept. 5. The left-hander struck [out] [K] [six] [Q]

Figure 22: Examples of high attention in examples in OpenWebText-10K for head 6.7 in GPT-2

**Composition Scores Between Max Component of Every Head and 9.9 Query (Mover Head)**

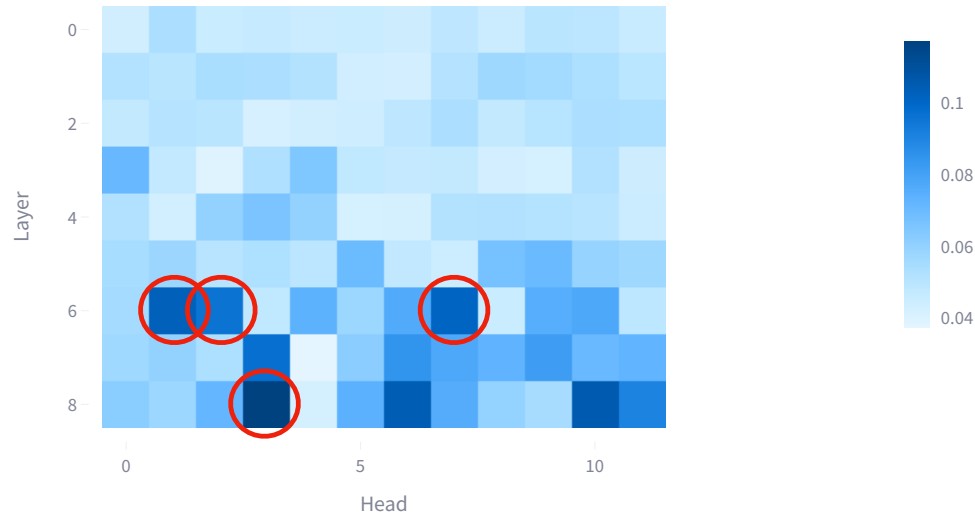

Figure 23: We look at the top scoring component matrices in every head before layer 9 to attention head 9.9. We find several heads not previously identified in circuit analysis that have strong communication to it. These are attention heads 6.1.1, 6.2.5, 6.7.3, and 8.3.0. 8.3.0 in particular, has the strongest composition with 9.9 out of any head.

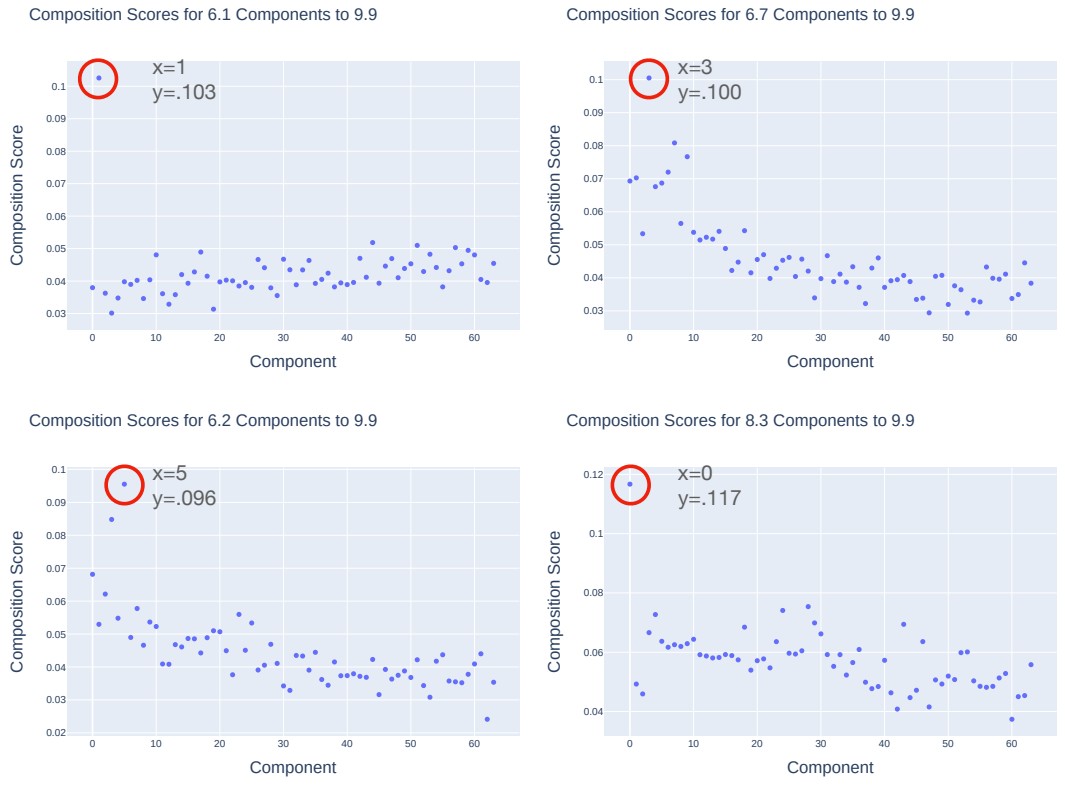

Figure 24: Per component composition scores for the 4 identified heads communicating with 9.9. Each has an outlier component that communicates most strongly with 9.9's QK circuit.

## Mean Attention for Likely and Unlikely Nouns 8.3

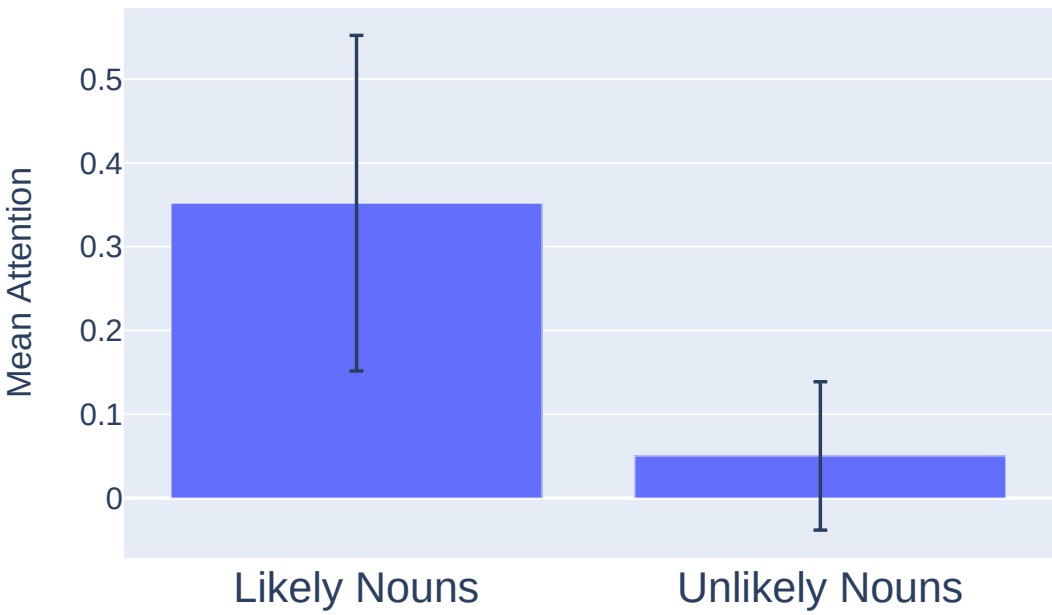

Figure 25: When predicting the continuation of sentences like 'I saw the book. I am reading the', head 8.3 is attends much more strongly to the given noun (book) when the verb is appropriate vs. inappropriate (e.g., eating). The average attention score is around 35% for likely nouns and around 5% for unlikely nouns.

## Attention Scores for 9.9 with and without Steering 8.3.0

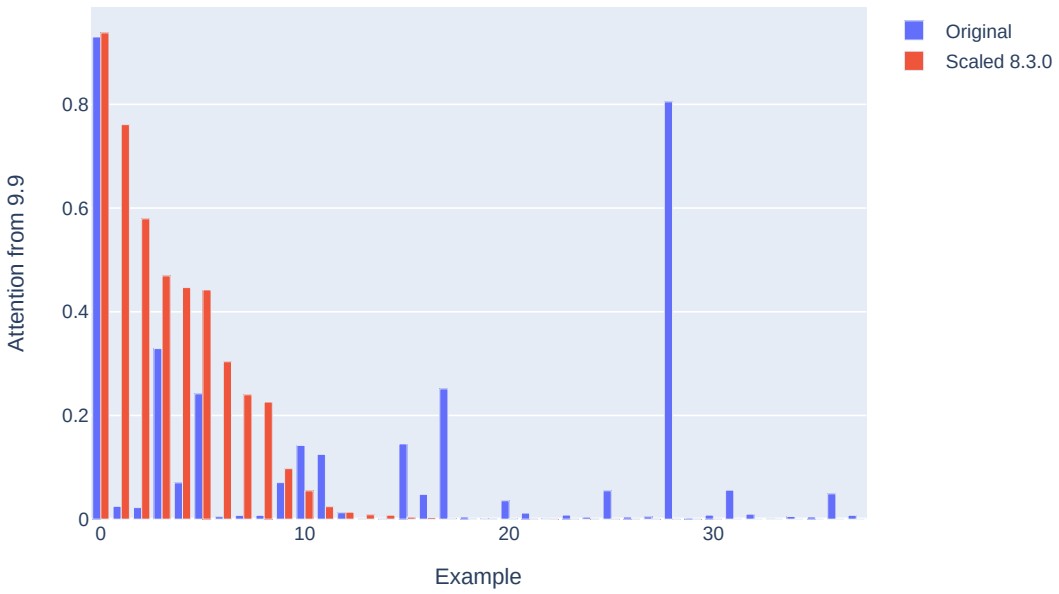

Figure 26: Interventions on 8.3.0 cause an increase in the attention of 9.9 onto the token 8.3 attends, although inconsistently. The x-axis shows results for individual examples. This is an average increase from 9% to 12%.

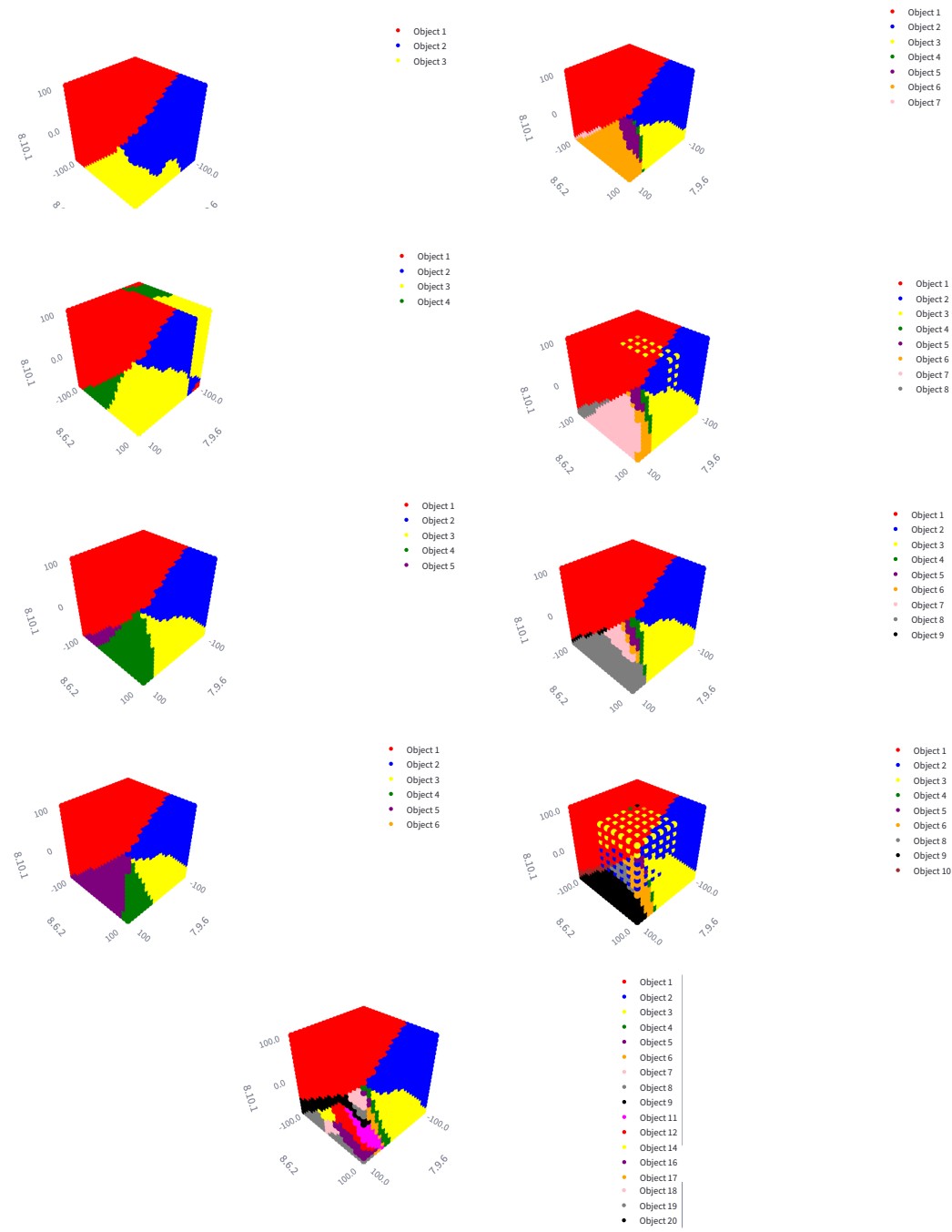

Figure 27: How the 3D inhibition subspace responds to a different number of objects in laundry list prompts. As we add objects, a new 'slice' of the space is allocated (not always visible) for attention to that object until the middle set of objects is squeezed into a small neighborhood of the space. The space is very well structured, except for two cases where artifacts form in the 8 and 10 object settings.

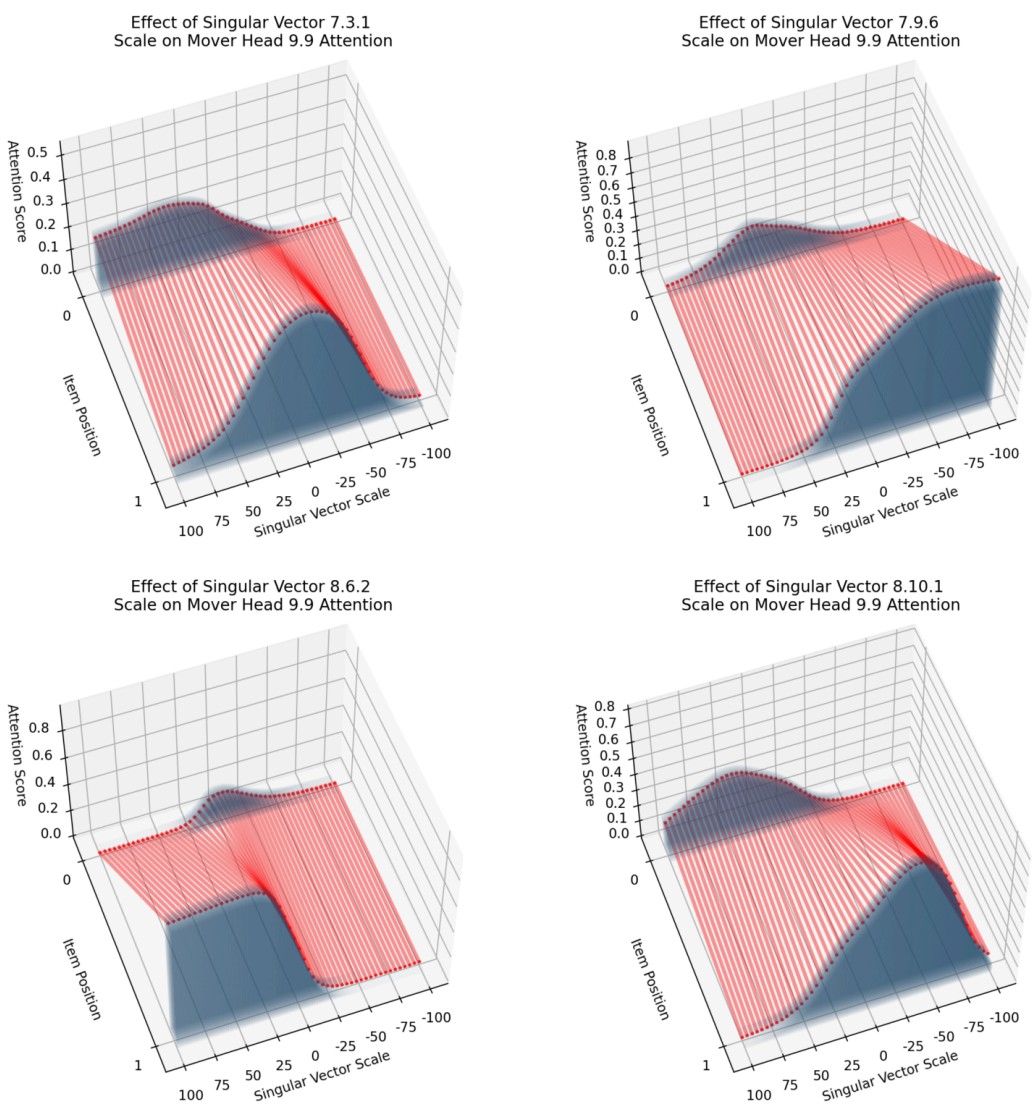

Figure 28: 2 Objects

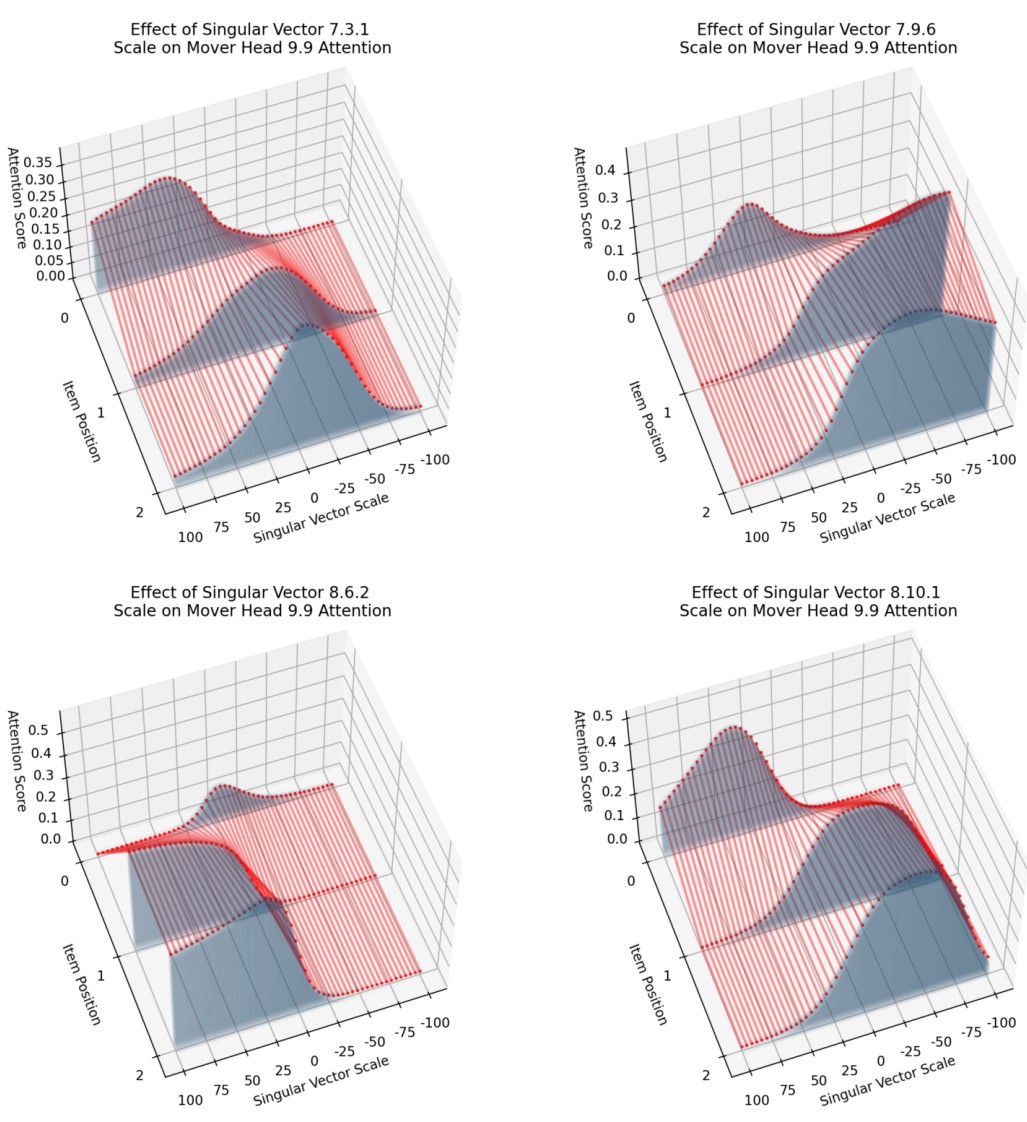

Figure 29: 3 Objects

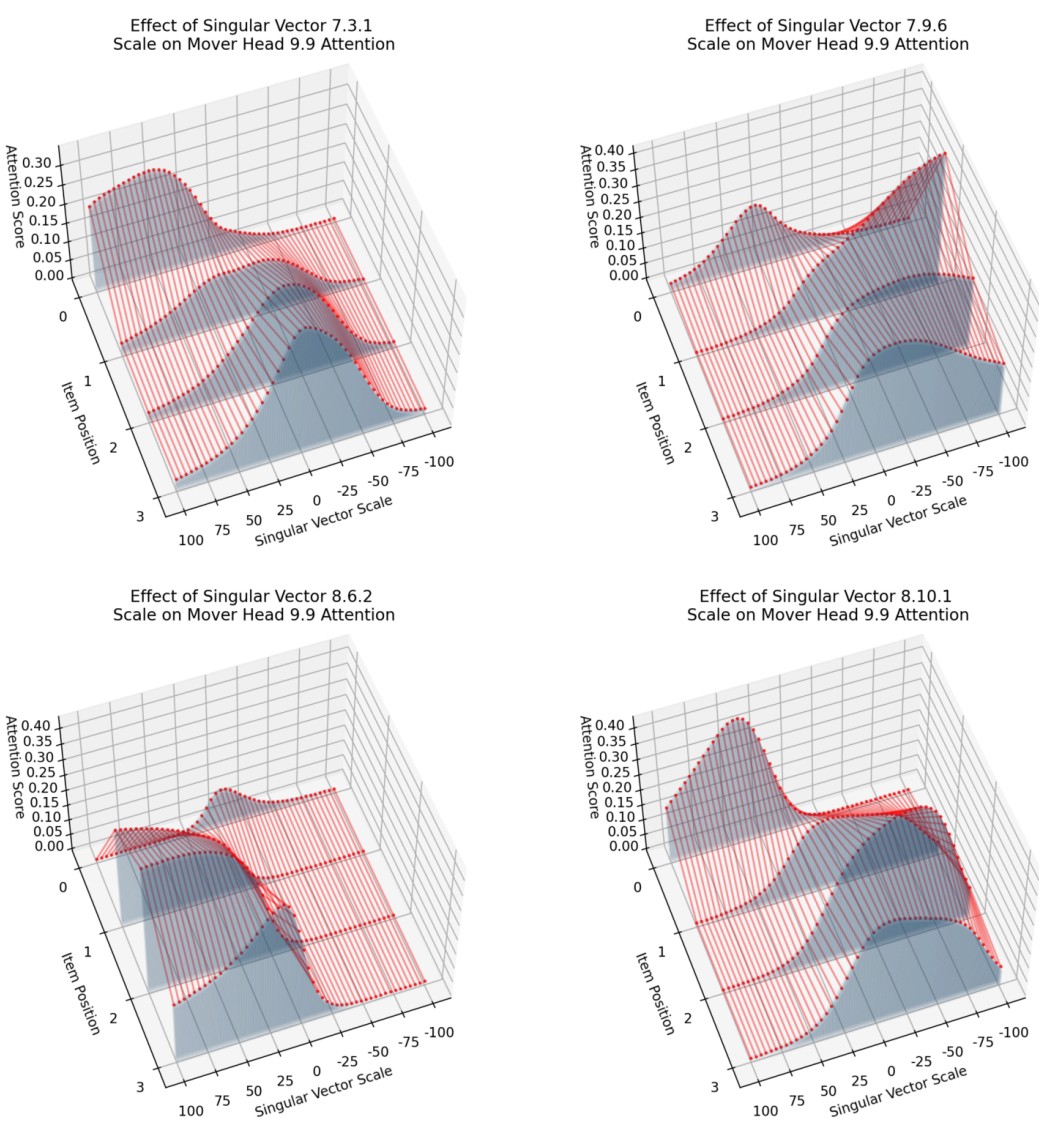

Figure 30: 4 Objects

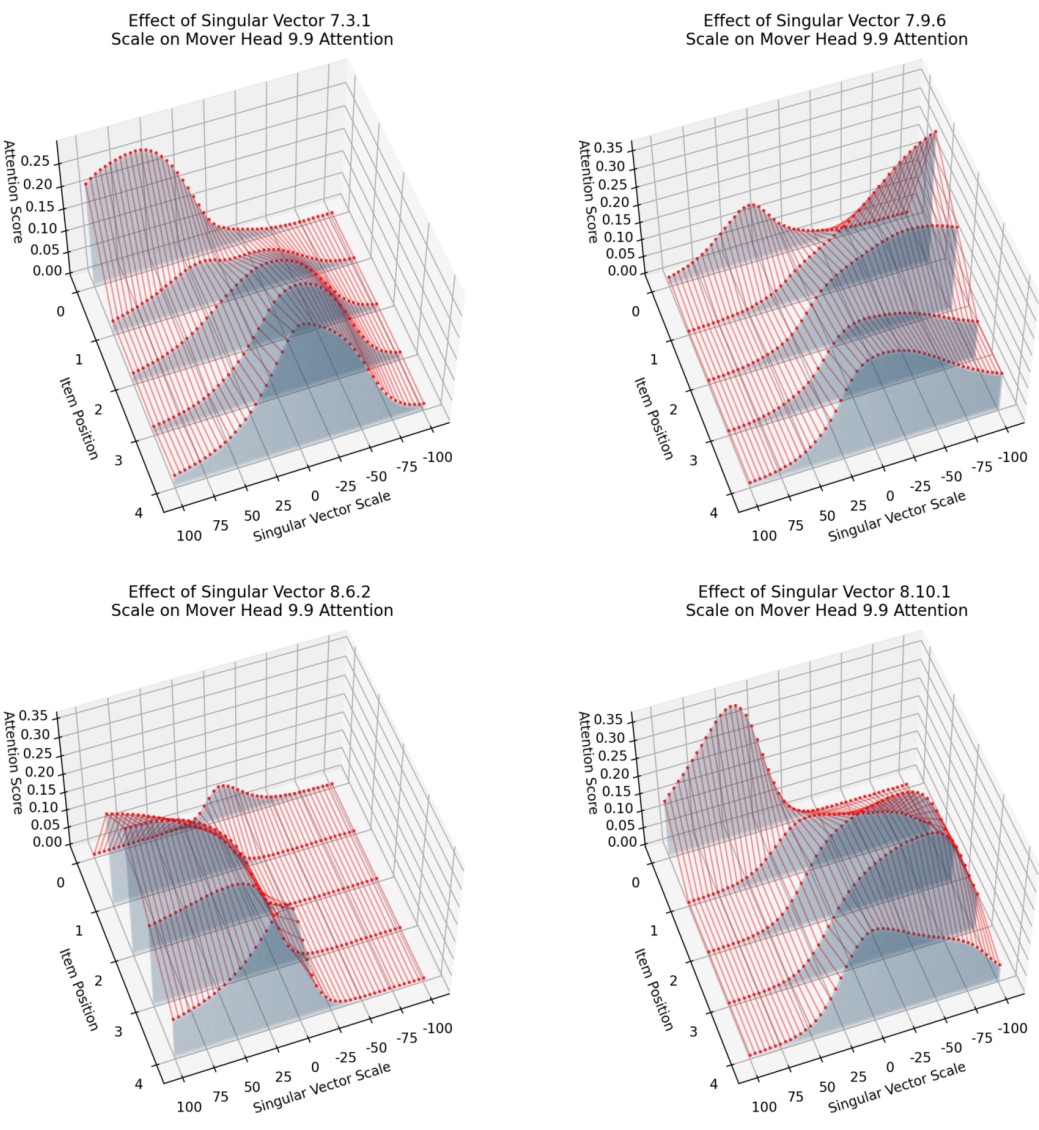

Figure 31: 5 Objects

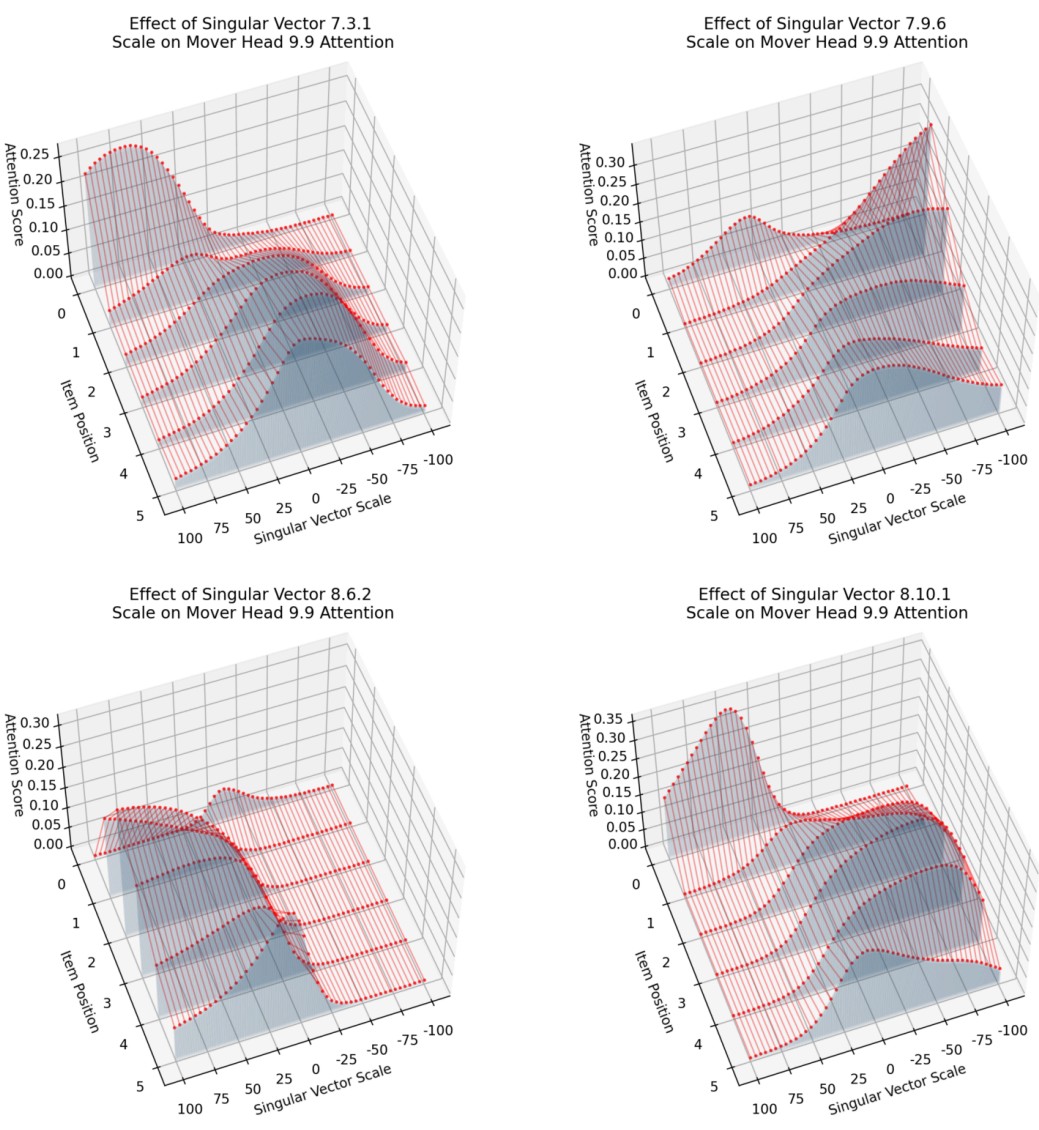

Figure 32: 6 Objects

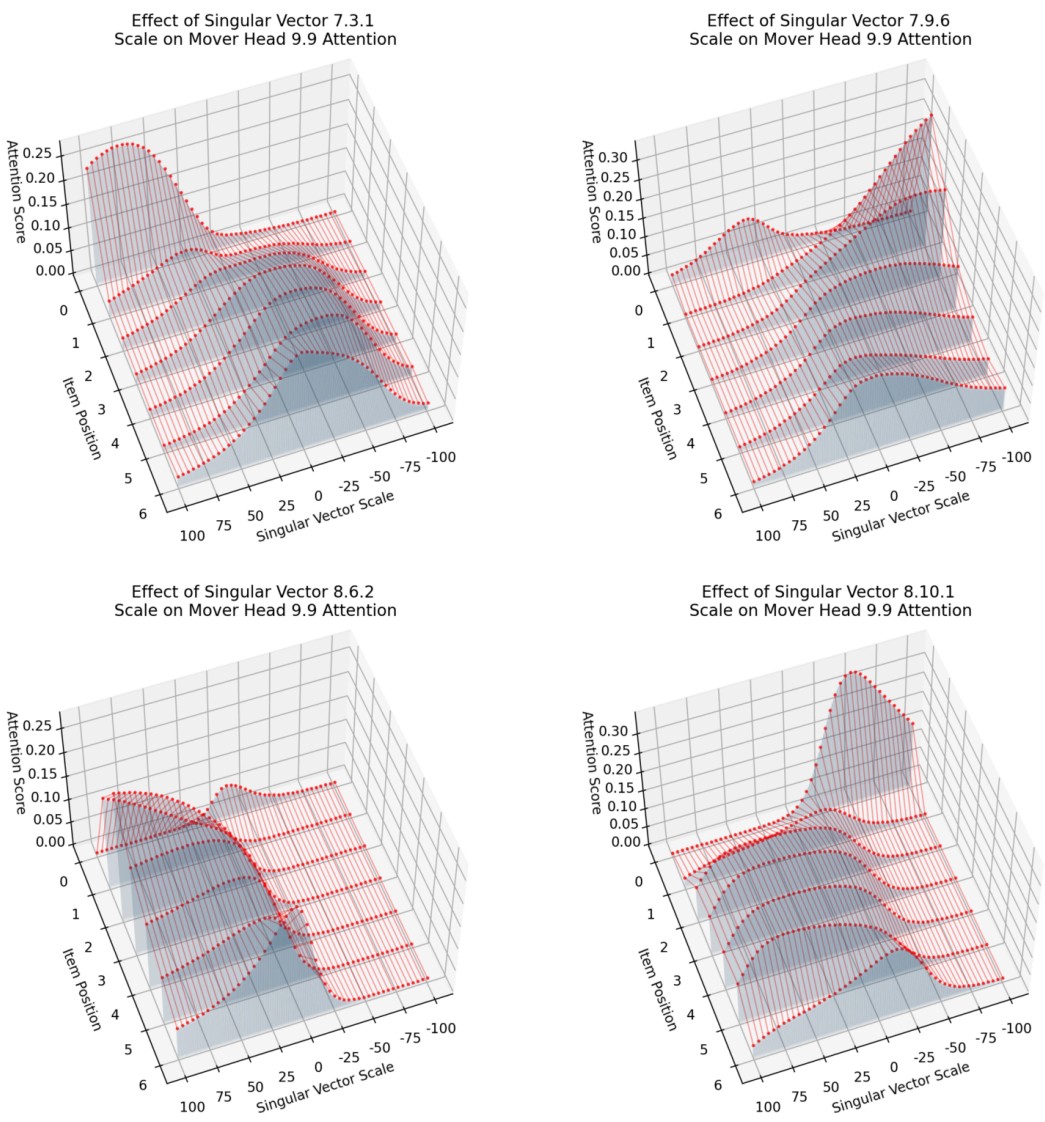

Figure 33: 7 Objects

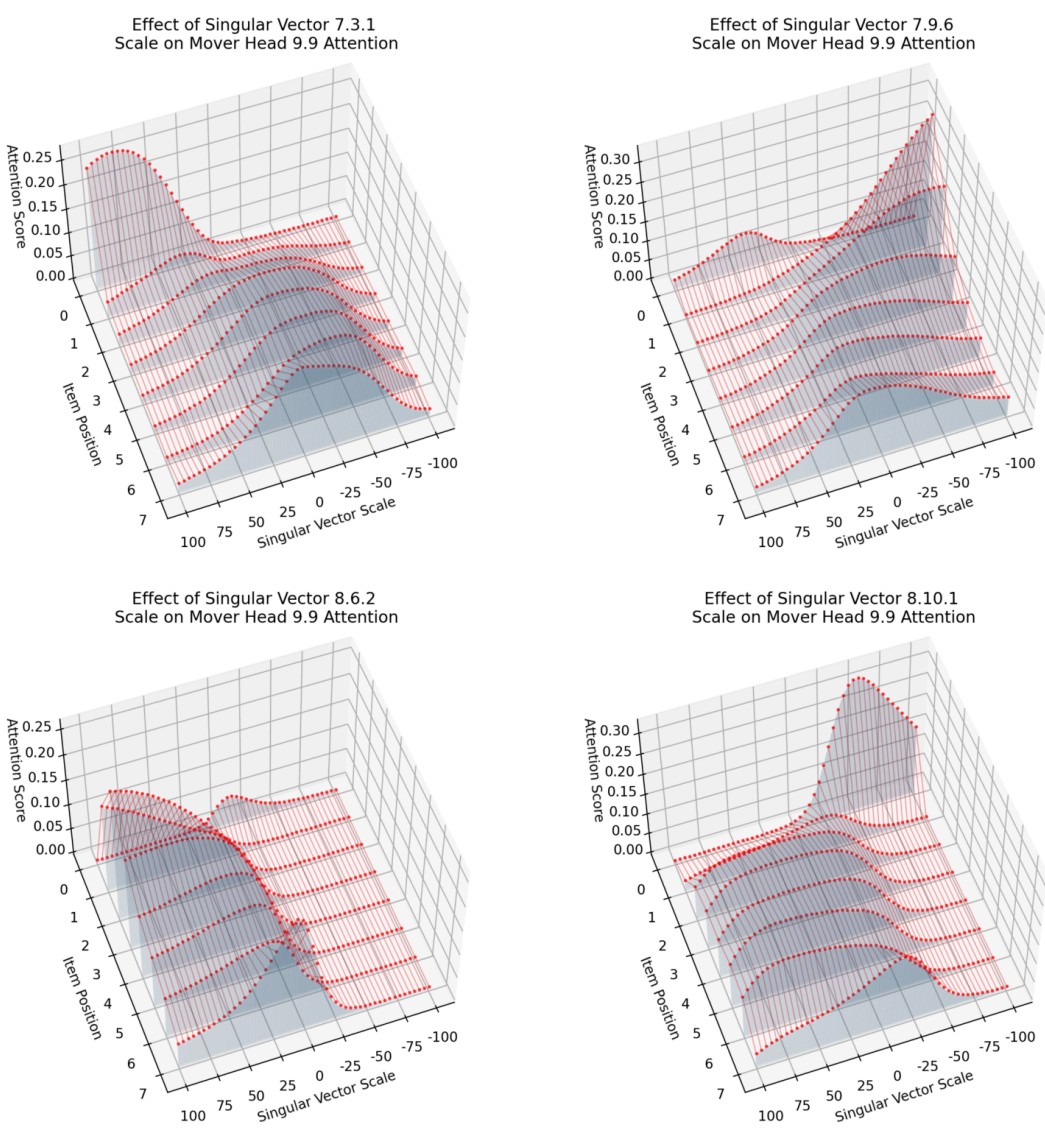

Figure 34: 8 Objects

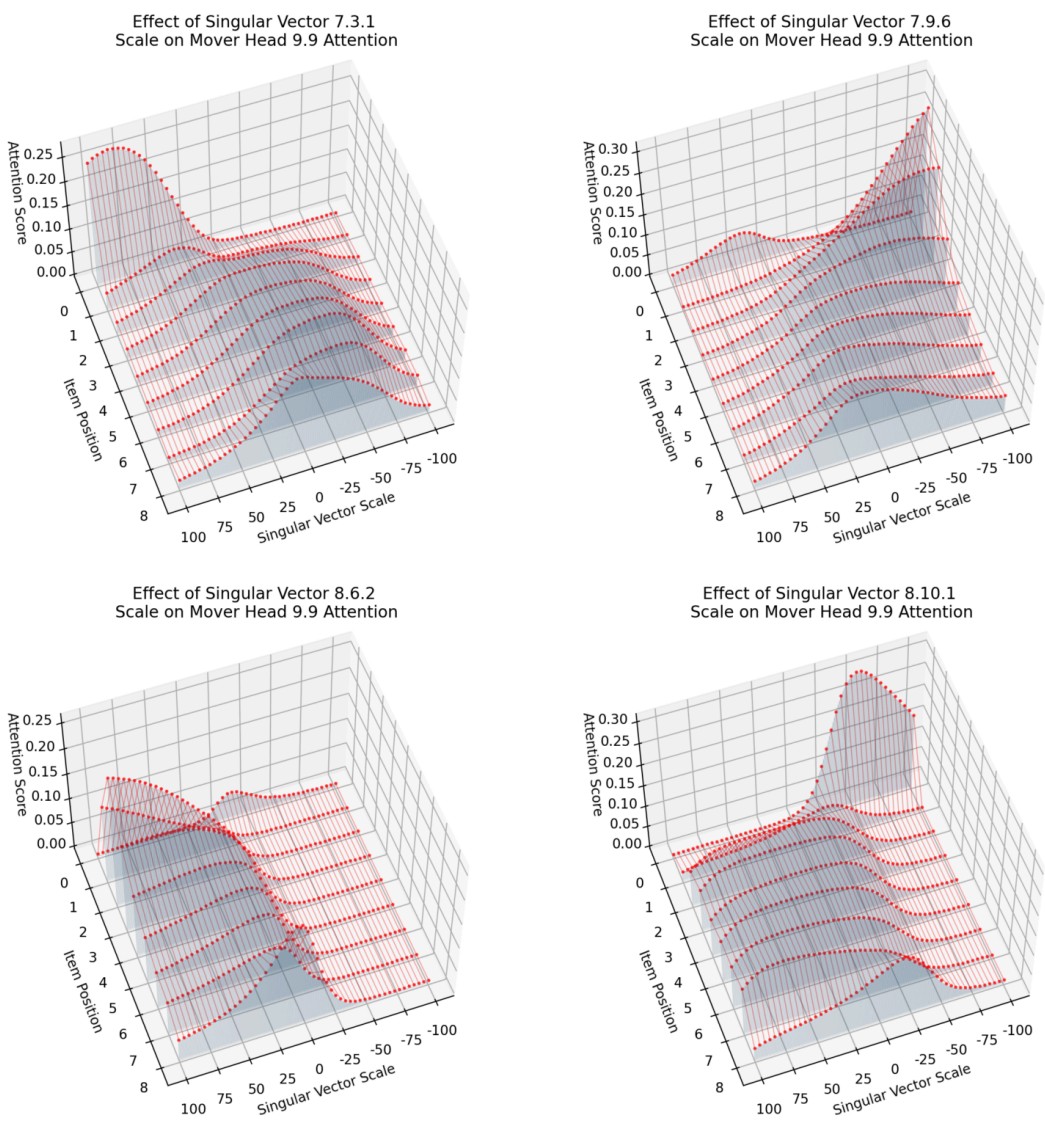

Figure 35: 9 Objects

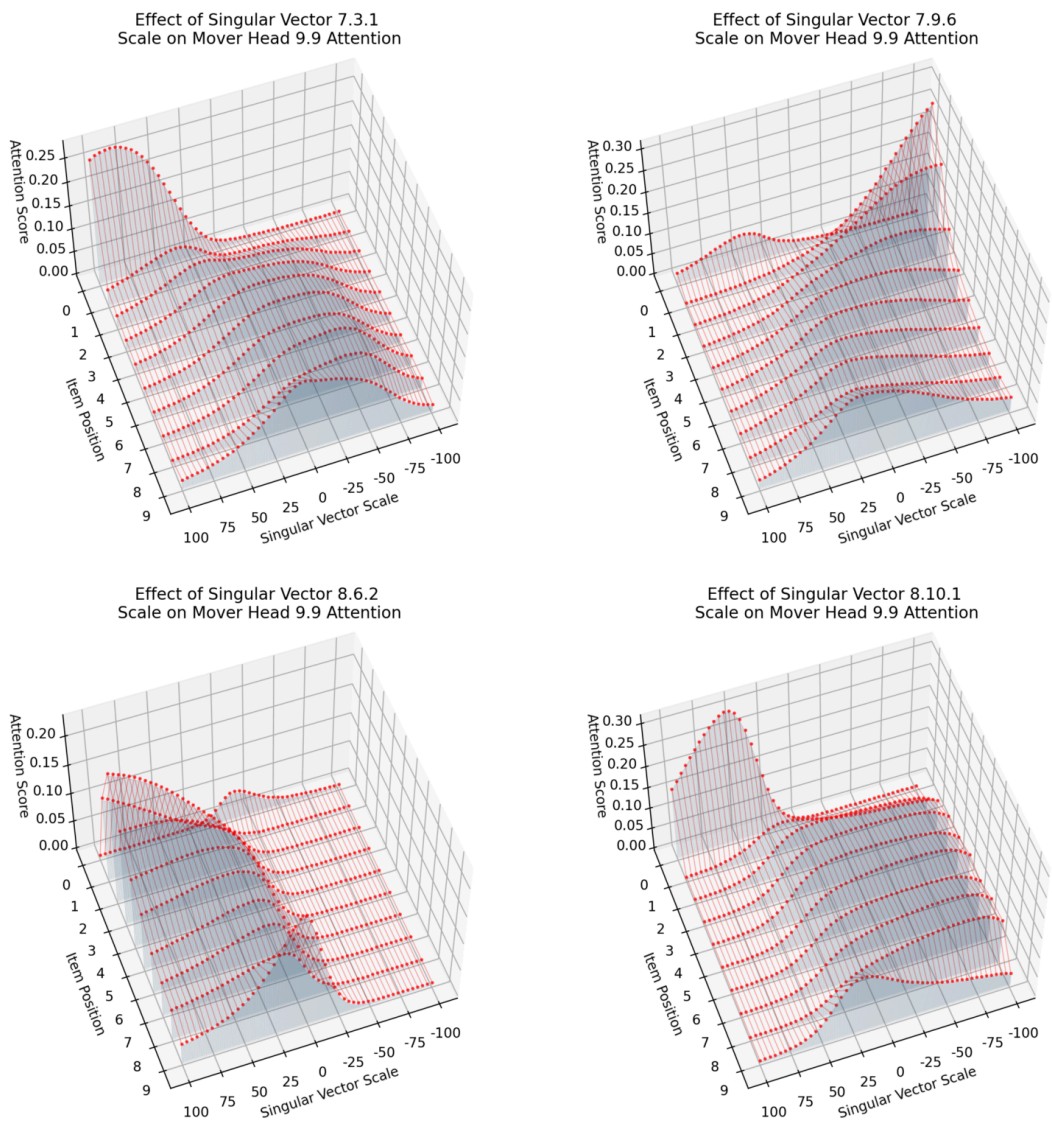

Figure 36: 10 Objects

## L Compute

The models we use in this paper are small, in the range of 100M parameter tranfsormer models. The compute required to reproduce the results is therefore relatively small, but does require access to modern GPUs. We primarily used Nvidia 3090 GPUs for this work. Running the linear combinations of inhibition components in Section 5 was the most expensive experiment. Each dataset took about 12 hours on either a RTX 3090 or Quadro RTX gpu.

