# OpenReview forum: "Talking Heads: Understanding Inter-Layer Communication in Transformer Language Models"
_NeurIPS.cc/2024/Conference — NeurIPS 2024 poster_

### Official Review · Reviewer_7Dor · 2024-06-28

**Soundness:** 3
**Presentation:** 3
**Contribution:** 4
**Rating:** 6
**Confidence:** 4

**Summary:**

This paper investigates the interaction between attention heads at different layers in a transformer. They primarily study the “inhibition-mover subcircuit”, a previously identified attention head interaction from circuit analysis work [1,2]. They show the interaction between heads can be characterized by a low-rank subspace, and show it is helpful in reducing prompt sensitivity on a simple item-recall task.

[1] Wang, et al. Interpretability in the Wild: a Circuit for Indirect Object Identification in GPT-2 small. 2023. (https://openreview.net/forum?id=NpsVSN6o4ul)

[2] Merullo, et al. Circuit Component Reuse Across Tasks in Transformer Language Models. 2024 (https://openreview.net/forum?id=fpoAYV6Wsk)

**Strengths:**

- The work is well-aware of existing related literature, and the questions posed are interesting.
- The work is well-motivated - current LLMs are often not robust to prompt variations, and understanding and mitigating the reasons for this is of value.
- The experiments demonstrate that the identified low-rank subspace can be used to effectively improve model performance on the item-recall task they introduce.

**Weaknesses:**

- The study is limited to fairly small transformers (GPT2-small, and Pythia-160m in appendix), and it’s hard to know whether the results will generalize to more complicated tasks or methods will scale to larger models. Additionally, It would also be helpful to know whether larger models also struggle with the item-recall task (and whether they have similar inhibition-mover subcircuits).
- The composition score presented in Equation 1 should be presented in context (i.e. seems to have come from Elhage, et al. [3]? Is there earlier use of this score elsewhere?). Besides substituting individual SVD components in for QK or OV matrices, are there any other changes compared to how [3] measure the composition score?
- It is unclear to me whether the failure mode of GPT2 small on the item-recall task is due to (a) not correctly identifying duplicated objects, or (b) not suppressing correctly identified duplicated objects when going to copy. This is because Figure 4 shows the attention pattern of mover heads can be influenced by either duplicate token head channels or inhibition head channels. Is this because the duplicate token channel influences the inhibition signal? (i.e. its effect on the mover head is mediated by the inhibition head?)
- There are some portions of the paper that present results without experimental details. One example is on Lines 228-231: "On OpenWebText ... we find inhibition heads are primarily active in lists and settings where repetition...". Providing details about the experiments run to validate this claim would help strengthen the argument made. In addition, what does it mean for an attention head to be "active" on text or not? Is this measured by its attention pattern or something else?


[3] Elhage, et al. A Mathematical Framework for Transformer Circuits. 2021 (https://transformer-circuits.pub/2021/framework/index.html)
___
There are a few places with incomplete sentences or minor typos. I've listed a few I noticed below:

- Line 174-175 - “... test how this affects.”, (incomplete sentence)
- Line 187 - “communicatoin” channels
- Line 214 - “non”
- Line 276 - “repersent”
- Line 290 - “featuers”
- Line 506 - "These results are in", (incomplete sentence)

**Questions:**

- When you say a head is “highly active” on a prompt or task [e.g. Line 229, 255], what is meant by that? Is this measured by its attention pattern or something else? When would an attention head be "non-active"?

- When using a subspace to steer the inhibition score, the singular values needed to flip attention from IO to S (or vice versa) seem rather large. What are the typical ranges of the singular values you see in your decomposition of the weight matrices you’re investigating?

- As one of the other well-studied/well-known subcircuits, have you done any analysis to understand whether the induction subcircuit is also dominated by a low-rank subspace? Or is this specific to the inhibition-mover circuit you study?

- In section 5, how does trying to learn optimal singular vector weightings either via gradient descent or regression compare to doing a grid search over 3-d points?

**Limitations:**

The authors have addressed limitations of their work, which are reasonable.

---

> ### Author Rebuttal · Authors · 2024-08-07
>
> > The study is limited to fairly small transformers (GPT2-small, and Pythia-160m in appendix), and it’s hard to know whether the results will generalize to more complicated tasks or methods will scale to larger models. Additionally, It would also be helpful to know whether larger models also struggle with the item-recall task (and whether they have similar inhibition-mover subcircuits).
>
> Thank you for raising this. Please see our rebuttal which primarily focuses on this point. In summary, yes larger models struggle with this task despite the simplicity and we argue this is an isolated mechanism that is part of a very basic and general language modeling mechanism learned by the model.
>
> > The composition score presented in Equation 1 should be presented in context (i.e. seems to have come from Elhage, et al. [3]? Is there earlier use of this score elsewhere?). Besides substituting individual SVD components in for QK or OV matrices, are there any other changes compared to how [3] measure the composition score?
>
> As far as we know, the composition score does not appear anywhere before Elhage et al. And that is correct, this iso our only modification to get it to work
>
> > It is unclear to me whether the failure mode of GPT2 small on the item-recall task is due to (a) not correctly identifying duplicated objects, or (b) not suppressing correctly identified duplicated objects when going to copy. This is because Figure 4 shows the attention pattern of mover heads can be influenced by either duplicate token head channels or inhibition head channels. Is this because the duplicate token channel influences the inhibition signal? (i.e. its effect on the mover head is mediated by the inhibition head?)
>
> The reviewer is correct in their last statement, it is because the duplicate token head affects the inhibition head, which affects the mover head. This is established in the circuit identified by [Wang et al., 2023](https://arxiv.org/abs/2211.00593). If this is still confusing to the reviewer we can go into more detail. Also to be clear, when we do the edit here, we are only editing the duplicate token subspace at the point the inhibition head sees it, so it is following the logic of duplicate token head-->inhibition head-->mover and **not** duplicate token head-->mover head (where --> is read as 'affects'/'changes'). We expect a lot of familiarity with previous work, which is not entirely fair to all readers, so to remedy this we added a section to the appendix outlining the entire IOI circuit in GPT2 from that paper (as well as the circuit we find for IOI on pythia).
>
> > When you say a head is “highly active” on a prompt or task [e.g. Line 229, 255], what is meant by that? Is this measured by its attention pattern or something else? When would an attention head be "non-active"?
>
> Yes this is measured by the attention pattern. We apologize for the lack of details, please see the rebuttal for an operationalization of these terms and how this analysis was carried out. They will be included in the final draft.
>
> > As one of the other well-studied/well-known subcircuits, have you done any analysis to understand whether the induction subcircuit is also dominated by a low-rank subspace? Or is this specific to the inhibition-mover circuit you study?
>
> We also looked at the induction head in Figure 6 (left) and Appendix H and found that the one that we looked at was not low rank, so not every head composition is like this. However this is not specific to the inhibition-mover/duplicate-inhibition subcircuits because we found plenty of other compositions between random heads that were similarly sparse. Unfortunately, we didn't have space to also analyze those and assign function to them because they are not part of any known circuits (like the IOI components). This is a very interesting direction for future work. We'd be happy to discuss more with the reviewer about what information we could/should include regarding this, though.
>
> > In section 5, how does trying to learn optimal singular vector weightings either via gradient descent or regression compare to doing a grid search over 3-d points?
>
> This is a very cool idea that we did not consider when writing/have time for in the rebuttal. We kept the 3-d points analysis extremely basic. Would this involve optimizing directly in that 3d space? Please re-raise this point during the discussion if the reviewer finds this an interesting addition to the paper.
>
> We hope we have resolved all of the reviewer's concerns which we believe we properly addressed in the rebuttal document or here, by clarifying details from the paper.

---

> > ### Comment · Reviewer_7Dor · 2024-08-08
> > **Thank you for your reply**
> >
> > I have read your response, as well as the other reviews. Thank you for the additional information regarding results on larger models and your viewpoint on how the simple recall task may relate to more general language modeling capabilities. I do think the other reviewers have brought up some valid concerns, (e.g. limited related work discussion, generality of the findings), but I think the authors have done an adequate job of trying to address most of them. I think the authors’ proposed changes and clarifications (along with cleaning up of the typos/ and general presentation) will strengthen the paper. As it stands, I think the paper would be an interesting contribution to the conference and would like to keep my score.

---

> ### Author Response · Authors · 2024-08-12
> **Thank you for the reply**
>
> Thank you for considering the rebuttal and other reviews, and suggestion to accept. We agree with the remaining points on related work and will make sure adequate space is provided in the camera ready.

---

### Official Review · Reviewer_2UBx · 2024-07-06

**Soundness:** 3
**Presentation:** 3
**Contribution:** 3
**Rating:** 7
**Confidence:** 4

**Summary:**

The paper investigates the communication between attention heads across different layers in Transformer-based language models. First, it establishes that a previous composition metric, which has been shown to be useful in toy settings, is noisy in larger models when tested on the Indirect Object Identification (IOI) task with a known circuit. The authors then propose a modification using Singular Value Decomposition (SVD), rewriting the original matrix as the sum of outer products of the left and right singular vectors, scaled by the corresponding singular values. This new metrics reveals inter-layer communication more clearly and provides a less noisy signal. This is causally verified by zeroing out a single singular value on the IOI task which reduces the inhibition score notably.

The authors then study these specific communication channels in detail, showing how interventions in these channels can affect the model's downstream behaviour. Interestingly, they demonstrate that this mechanism is independent of the specific context, functioning as a general pointer over lists. Finally, the authors use these insights to study the seemingly arbitrary sensitivity of language models to the order of items in lists using an artificial laundry list task. They show that as the list length grows, the internal structure in the communication partly collapses and makes it hard for the model to select the correct item. This can in part be addresses using a custom intervention, again demonstrating the causality of this phenomenon on the model behaviour.

**Strengths:**

- This paper addresses a fundamental question in interpretability of transformer-based language models, namely how attention heads selectively communicate with each other. Although it was known from prior work that attention heads compose with heads in other layers, to the best of my knowledge, this is the first in-depth study of inter-layer communication in large language models.
- The authors use a variety of techniques to systematically test and causally verify their hypotheses; I also find the use weight-based decompositions instead of activation-based techniques intriguing, as it enables static analysis of neural networks without the need for input data.
- The findings are connected to previous observations on language model robustness, explaining a seemingly arbitrary sensitivity of language models to the order of items in lists.

**Weaknesses:**

- The proposed composition score does not work for models using relative positional embeddings like RoPE, which are standard in state-of-the-art language models such as Gemma [1]. This makes the composition score not directly transferable.
- The results could be significantly strengthened if the technique was used to conduct an unsupervised search for head compositions within the model, rather than focusing solely on the IOI task with a known circuit.

Minor issues:
- Typo in line 11: “via analysis *of* their weight matrices”
- Typo in line 176: “the OV [or] matrix”
- Typo in line 187: “communicatoin”
- Overlapping image in Fig. 5, covering part of the “10"

[1] G. Team et al., ‘Gemma: Open Models Based on Gemini Research and Technology’, arXiv [cs.CL]. 2024.

**Questions:**

1. Could you elaborate on the intervention used to improve model performance on the laundry list task? Specifically, what operations do you perform to "set the model components in a certain area of the 3D space"?
2. Can the proposed composition score be used to identify compositions of attention heads without prior hypotheses?

**Limitations:**

I believe the limitations are properly addressed, although the most important limitation - that the composition score is not directly applicable to models with relative positional embeddings - is somewhat hidden in the appendix.

---

> ### Author Rebuttal · Authors · 2024-08-07
>
> Thank you for the review. We are happy that the reviewer enjoyed the paper and we appreciate the encouraging comments regarding static weight analysis, we think there is a lot of interesting work to be done in this area.
>
>
> > The proposed composition score does not work for models using relative positional embeddings like RoPE,
>
> We'd like to point out that this is only for query/key composition and does not occur for composition between values and outputs. Pythia uses RoPE and we cover positive results on this model using composition in the appendix. We use the composition score to do static circuit analysis on part of the Pythia IOI circuit, which was not previously known before (which we later validate with path patching). See the below point. See also the rebuttal for more info.
>
> > Can the proposed composition score be used to identify compositions of attention heads without prior hypotheses?
>
> Yes absolutely, the compositions can be taken between pairs of heads, and outliers can be searched for. From there, new connections can be found and later analyzed on data. We had to do IOI circuit analysis on pythia to find inhibition heads, and the composition score was used for part of this. After starting the circuit analysis and finding the inhibition heads, the value composition score revealed the induction heads that connect the circuit, which path patching later confirmed
>
> > The results could be significantly strengthened if the technique was used to conduct an unsupervised search for head compositions within the model, rather than focusing solely on the IOI task with a known circuit.
>
> Thank you for raising this. We definitely hear the reviewer on this, but we're not sure how we could fit this in without it disappearing in the appendix. We hope that the above point about our analysis on pythia without a known circuit is satisfactory for the scope of this paper.
>
> > Could you elaborate on the intervention used to improve model performance on the laundry list task? Specifically, what operations do you perform to "set the model components in a certain area of the 3D space"?
>
> Yes, in plain english, each inhibition head is responsible for one direction (we focus on three of them so that we can visualize the space without dimensionality reduction). We overwrite the output of each head to be some coordinate in its corresponding direction. That is, some scalar times the unit vector corresponding to the inhibition component we previously identified. A very important part of this intervention is that it makes it impossible for the head to attend to any of the previous context (the Q*K operation becomes unused), which is why we are able to say it is content independent: we control which item in the context is being attended to/ignored without letting those heads attend to them

---

> > ### Comment · Reviewer_2UBx · 2024-08-11
> >
> > Thanks for clarifying the intervention details and confusion regarding the applicability to RoPE-based models. I also agree that there is no space in the main paper to perform and discuss unsupervised searches for head compositions and would thus suggest to leave it for future work. I think that other reviewers have raised some valid concerns (e.g. generality of findings, relatively unspecific title), but I still believe it is an interesting, and valuable case study with potentially broader applicability, and that most concerns could be addressed in the camera-ready version. Overall, I would like to see the paper being presented at the conference and thus suggest accepting it.

---

> > > ### Author Response · Authors · 2024-08-12
> > > **Thank you for the reply**
> > >
> > > We are glad the clarification was helpful. Thank you for considering our rebuttal and for your suggestion to accept

---

### Official Review · Reviewer_erzS · 2024-07-11

**Soundness:** 2
**Presentation:** 3
**Contribution:** 3
**Rating:** 6
**Confidence:** 4

**Summary:**

This paper explores the routes and mechanisms of information transfer between heads of transformer large language model. They hypothesize the presence of low-ranking communication channels between attention heads from different layers within residual connections. The authors propose a method based on Singular Value Decomposition of weight matrices to detect those channels in a pretrained model; this method doesn’t require any additional data. They show that this method can uncover intricate and interpretable intrinsic structures of a transformer, which in turn can be manipulated to targetly adjust the model’s performance. The authors illustrate it by significantly improving the performance of GPT-2 on the list item recall task.

**Strengths:**

•) Very interesting and novel idea.

•) Good analysis; promising results that can facilitate further progress in interpretability and understanding of the innerworkings of large language models.

•) The article is well-written, and the text is easy to follow. The figures are nice and illustrative.

**Weaknesses:**

1) The proposed method of Communication Channel Interventions improves model performance on a synthetic task (Laundry List); however, it is not clarified how such modification will affect model's performance on other tasks.

2) The contextualization in the research field (Section 6, Related Work) is short and covers rather little information on previous works about information passage within transformer models and attention mechanism.

3) The limitations of the method do not allow for a straightforward implementation on newer models with relative positional embeddings.

4) [Minor] The possible practical applications of the achieved results are outlined vaguely.

**Questions:**

1) You identify several types of attention heads: mover, duplicator, and inhibitor, but could a single head exhibit traits of multiple types? Also, is this classification not exhaustive (i.e., there are heads that do not fall into any of the aforementioned categories)?

2) If there are two interventions and each of them is beneficial (i.e., increases the model's performance) on its own, how often is their composition (i.e., applying them simultaneously) beneficial? Is it possible to somehow predict such pairs of interventions without directly computing the quality of each pair?

3) How many heads were affected by the interventions in your experiments in Section 5? Do you have information on how the increase in performance depends on the number of modified heads?

**Limitations:**

Authors clearly outline the limitations of their work in the dedicated section (in Appendix B).

---

> ### Author Rebuttal · Authors · 2024-08-07
>
> Thank you for the review. We are very happy to hear that the reviewer found the results interesting, promising for facilitating future work, and the paper easy to follow. The reviewer had some concrete questions and concerns about the generalizability of the method to new models and results to new tasks. We believe we have satisfactory answers to these below:
>
> >  The limitations of the method do not allow for a straightforward implementation on newer models with relative positional embeddings.
>
> We would like to point out that this is only for composition with queries and keys in models with RoPE embeddings, in case it wasn't clear. Value-Output composition is entirely unaffected, and we have results with Pythia (a RoPE model) in Figure 12. This is certainly still a limitation, but we believe in the future it will be surmountable. In addition, composition with mlps (mlp to attn, attn to mlp, and mlp to mlp) is still a valid path for future work that we didn’t explore here, so we don’t think that our method will be limited to only certain kinds of models. Given the complexity of the mechanisms involved already, we figured it would be too much to find a satisfactory workaround to the query/key problem, work with MLPs, and provide our current analysis all in one paper. We'd also like to point the reviewer to the rebuttal which addresses some related points.
>
> > The proposed method of Communication Channel Interventions improves model performance on a synthetic task (Laundry List); however, it is not clarified how such modification will affect model's performance on other tasks.
>
> We would argue that the mechanism we uncover is an extremely general component in broader language modeling capabilities and possibly affects most tasks that involve some form of recall. Our evidence for this is the activity on both the IOI and Laundry list tasks (which would seem to coincidental otherwise), the content independence of the mechanism (see Line 281), and the consistent general activity on open domain text (see rebuttal document). In our rebuttal document we outline some evidence for this idea
>
> We will use the rest of the space to answer other questions:
>
> > How many heads were affected by the interventions in your experiments in Section 5? Do you have information on how the increase in performance depends on the number of modified heads?
>
> We use three of the four inhibition heads (7.9, 8.6, 8.10). We chose to leave out the fourth (7.3) because we wanted to be able to plot the results in 3D without dimensionality reduction (Figure 5) since that often can’t be done. As for why we chose 7.3: per Line 182: “changing 7.3 does not have a strong effect on its own” (see Figure 2).
> We saw some marginal increase by also including 7.3 but it was computationally expensive to run the full sweep. Based on the evidence we do have, the scores would probably be at baseline if we performed an analogous intervention on three random heads, though.
>
> > You identify several types of attention heads: mover, duplicator, and inhibitor, but could a single head exhibit traits of multiple types? Also, is this classification not exhaustive (i.e., there are heads that do not fall into any of the aforementioned categories)?
>
> We did not find, but don’t rule out that a single head could exhibit traits of multiple types. This classification is not exhaustive, and we do briefly study induction heads in the appendix. Still, we are excited by recent and evolving work  in characterizing other types of observed specializations in heads (e.g., [successor heads](https://arxiv.org/abs/2312.09230)).

---

> > ### Comment · Reviewer_erzS · 2024-08-13
> >
> > Thank you for answering my questions and for clarifications provided; I have also read other reviews and responses to them.
> >
> > I would suggest you somehow highlight in the main text the fact that some of the detected phenomena (Value-Output composition) are unaffected by addition of RoPE (and probably other methods of relative position encodings) to the model, because, in my opinion, this would greatly increase the actuality (and interestingness) of the presented findings to the readers.
> >
> > I still have some concerns regarding the `side effects' of Communication Channel Interventions. It may be exceeding the scope of this paper, but I would encourage you to evaluate the performance of the edited models at text generation (although it may be hard to define a formal setting), because it might so happen (this is an intuitional guess) that inhibition channels have an important role during the generation of larger texts.
> >
> > Finally, I think that the method proposed in the paper is quite interesting, and the reported results can be valuable for future research in the field. I think that changes proposed by authors will strengthen the paper, and I will update my ratings and raise the overall score.

---

> > > ### Author Response · Authors · 2024-08-13
> > >
> > > We are happy to hear the clarifications improved the perception of the paper. Thank you for the suggestions, we will definitely specify that RoPE does not affect Value composition.
> > >
> > > > It may be exceeding the scope of this paper, I would encourage you to evaluate the performance of the edited models at text generation (although it may be hard to define a formal setting)
> > >
> > > We can make this part of the appendix that describes the inhibition heads' role on open domain text, and use the examples we find as targets for interventions. Since there was some interest from reviewers in this kind of thing, we think it could be useful, and doesn't necessarily overextend the paper.
> > >
> > > Thank you again to the reviewer for their help and effort in improving this paper

---

### Official Review · Reviewer_Veos · 2024-07-13

**Soundness:** 2
**Presentation:** 1
**Contribution:** 3
**Rating:** 5
**Confidence:** 4

**Summary:**

Building upon prior work in Transformer circuits (Elhage et al., 2021; Wang et al., 2022), the paper identifies novel, low-rank communication channels between attention heads across layers via the Composition Score (a generalized cosine between the read-out weights of a lower-level layer and the read-in weights of an upper-level layer) between _low-rank_ factorizations of the weight matrices (obtained by SVD). The low-rank decomposition is critical, resolving the known issue of the vanilla Composition Score being very noisy.

Generally focusing on the role of inhibition heads, which are known to prevent the copying of certain tokens in the prompt, the paper identifies  a low-rank communication channel for inhibition and validates its role on a synthetic "Laundry List" task. The task requires recalling an item from a list of items specified in the prompt. It is shown that, when the identified (rank-1) inhibition channel is zeroed out (intervention), the model effectively stops passing the "inhibition signal." The role of the channel is validated for indirect object identification (IOI) and token indexing tasks in the GPT2-small model.

**Strengths:**

- Using the SVD as a remedy for fixing the noisy Composition Score is an intuitive and seemingly effective strategy.
- The paper’s main contributions, including the method and the new synthetic dataset, are reasonably well-motivated.
- The empirical results make sense (for the most part) and support the claim of low-rank communication channels for inhibition heads. It is interesting to see that the inhibition channels are content-independent (just pointing at names in the IOI task) or that it affects the accuracy on the laundry list task.

**Weaknesses:**

- Overall, while the paper is well-motivated and includes potentially interesting results, its presentation significantly hinders the reader from understanding the main takeaways of the paper.
    - First, a key piece of related work is the composition analysis of two-layer, attention-only models by Elhage et al. (2021), but the review of the relevant background work is simply too terse and unorganized. Terms like inhibition heads, inhibitor-mover subcircuit, value composition, and the QK/OV circuits (e.g., which weight matrices are we talking about exactly? why are they “low-rank”, especially when it’s not as low-rank as the subspaces we’re discussing here?), are not formally defined. Even things like why the composition score looks the way it is (why is it a matrix multiplication?) or the basic shapes of the weight matrices ($d$ by $d_h$ or $d_h$ by $d$?) involved are not written out in the “background” section. Without these in place, I think the paper is borderline incomprehensible to anyone outside the mech interp community.
    - Second, all the overclaims in the paper really hurt the overall message. The paper’s main thesis could simply have been something like “using the SVD in the Composition Score allows us to identify low-rank communication channels, as showcased by the identification of inhibition head channels,” which would be clear and interesting. But instead, the paper keeps trying to convey a broader, unsubstantiated message that their methods and experiments somehow are more novel and general than they should be.
        - “Understanding Inter-Layer Communication”: Exactly what is the scope of this inter-layer communication? Is it primarily/always from layer $l-1$ to $l$, or something more?
        - ”… using subspaces we call communication channels” (Intro): this terminology already existed in Elhage et al. (2021), but this phrasing makes it sound like the current paper came up with it.
        - “model components like attention heads” (what other components does this paper cover?)
        - “Three types of composition”: It appears that most of the focus in the paper, including the dataset itself, is on inhibition heads.
        - “Composition Score”: It should be clear to the reader that the score was first developed by Elhage et al. (2021), especially in Section 3.1. More generally, the intro should do a better job of clearly disambiguating what was introduced already in the literature and what this paper is adding to it.
    - Lastly, the writing is simply not well polished, and there are many typos and unnatural sentence structures that make it even more difficult to understand the overall paper. Just to name some of them:
        - Abstract, line 3 & line 11
        - Intro, line 34 (\citep)
        - Page 4, line 114
        - Page 4, line 128 (d \times d)
        - Figure 2 caption, second sentence
        - Page 5, lines 170—172 (broken parallelism)
        - Figure 3 caption, second sentence
        - Page 9, line 276

        There’s also a related issue of overusing informal analogies unnecessarily (attention heads “talking” to each other).

- Aside from the presentation, the biggest confusion for me is: generally, what implications do we have about inter-layer communication in Transformers from these findings? Besides inhibitions, what compositions do we expect to be represented as low-rank subspaces according to the paper’s approach? How far across layers can the channels be? Do we have any sense of what the rank tells us about the feature? The discussion section should be expanded substantially to at least mention or address some of these points.
- In terms of the proposed Laundry List task, a possible caveat is that models more recent (and larger) than GPT2-Small may actually do a good enough job of solving them. I am sympathetic to the fact that it is challenging to redo many of the analyses on larger models, but I think it is worth, at least, to test the accuracy (Fig. 1, left) on more recent, open-weight models, to see how much the motivation in the introduction is still relevant.
- The related work on SVD seems terse, knowing that SVD is arguably one of the most basic linear algebra tools in ML (any method based on PCA would also be relevant, for example).
- In the intervention experiments, the choices of the exact subspace (the number of components) and the intervention method (e.g., scaling diagonally for 2D) appear a bit heuristic and need better justification.

**Questions:**

- Are there other baseline methods to be considered beyond the vanilla and SVD-based Composition Scores for identifying communication channels?
- Page 6, lines 217--218: How exactly is the 2D duplicate token head found?
- Page 7, line 227: In what sense are these results “strong”? Is there a baseline?
- Page 7, lines 228--232: any quantitative results for this claim?
- Page 8, line 266: why is it top-3 this time? how should someone replicating your experiments choose this number?
- I think the talking heads analogy makes sense, but it’s one that was already used before by [Shazeer et al. (2020)](https://arxiv.org/abs/2003.02436) in a not-so-unrelated context (the paper is fully referenced in the circuits paper, for example). I would suggest changing the title or at least adding a footnote of disambiguation from that paper.

**Limitations:**

The paper states that it does not make claims about the different types of compositions across attention heads in different layers. Aside from that, the limitations are not well addressed; see weaknesses above.

---

> ### Author Rebuttal · Authors · 2024-08-07
>
> Thank you for the review. We are glad the reviewer found the methods interesting and well-motivated but disappointed the paper was difficult to read. This is a complicated topic that needs to fit in 9 pages, so it is difficult to catch everyone up and make a significant contribution in that time. Below, we answer questions, outline some proposed changes, and would also like to use this space to push back on the notion that we are overclaiming. After considering our points we would like to hear if the reviewer still feels this way/what they specifically feel is being overclaimed.
>
> >“Understanding Inter-Layer Communication”: Exactly what is the scope of this inter-layer communication? Is it primarily/always from layer l−1 to l, or something more?
>
> We do not find a preference for more recent layers. Please see Figure 2: heads in layer 7 communicate with the mover head in layer 9. The duplicate token head (3.0, see Figure 4) communicates with inhibition head 7.9 (a difference of four layers).
>
> > “using subspaces we call communication channels” (Intro): this terminology already existed in Elhage et al. (2021)
>
> We use this term to ground the paper to previously established literature but also provide a more specific definition rather than a notion of communication. Elhage et al., is extensively cited in this work, but this point is well taken because it was not our intention to appear to coin this term. This will be reframed in the camera ready to refer to this paper when introducing “communication channels”.
>
> > “QK/OV circuits (e.g., which weight matrices are we talking about exactly? why are they “low-rank””
>
> These are low-rank because of their shapes. For example, in one head, the value matrix projects from the embedding dimension (768) to the head dimension (64). E.g., the value matrix is 64x768 dims. This is therefore lower rank than the embedding dim. This has been consistent in the transformer architecture since Vaswani et al,. 2017 so we do expect some familiarity from a general reader.
> If the reviewer agrees it would be helpful, we would be more than happy to define these terms in a glossary in the appendix.
>
> Relatedly, we’d like to point out this isn’t a typo:
> >Page 4, line 128 (d \times d)
>
> We will add something like “V \in d x d_h and O \in d_h x d, so OV \in (d \times d)” to make this very clear
>
>
> > There’s also a related issue of overusing informal analogies unnecessarily (attention heads “talking” to each other).
>
> We tried to balance formalisms with intuitive analogies due to the complexity of the work. If the reviewer could, we would be interested in hearing which specific places more rigorous definitions would be helpful.
>
> > “all the overclaims in the paper really hurt the overall message”
>
> We are generally a bit confused about what the reviewer finds to be overclaimed. Perhaps the above points help clarify our work more. If not we would appreciate some specifics if the reviewer still feels this point is relevant after the rebuttal.
>
> > In terms of the proposed Laundry List task, a possible caveat is that models more recent (and larger) than GPT2-Small may actually do a good enough job of solving them.
>
> This is a great suggestion, please see our rebuttal doc. We tested multiple n-billion parameter models (up to 6.9B) from different model families and found that the larger/more recent models tend to get better but still struggle. We’d also like to point out some relevant related work:
> [Liu et al., 2023](https://arxiv.org/abs/2307.03172): highlights the broader inability of LMs to use information when it’s in the middle of a long context. This negatively affects retrieval augmented generation (RAG) systems, which are highly relevant to current SotA systems. We don’t claim this is a failure of the exact same mechanism, but it definitely connects our work to a substantial current problem.
>
> This point may also help answer the following:
>
> >”[a] confusion for me is: generally, what implications do we have about inter-layer communication in Transformers from these findings?”
>
> Our work points to a concrete capacity limit in inter-layer communication that leads to performance degradation on certain recall tasks. This work provides a promising avenue for understanding and improving “lost in the middle (Liu at al) type failures.
>
> > “Page 7, line 227: In what sense are these results “strong”? Is there a baseline?”
>
> Yes there is a baseline. In Makelov et al., 2023, they examine the IOI dataset. As a reminder, we report 97.5% and 35% with components directly taken from the weights. Their baseline achieves -8% FLDD 0.0% Interchange. By taking the gradient to directly optimize a single vector for this task they achieve 111.5% FLDD and 45.1% Interchange.
>
> > “In the intervention experiments, the choices of the exact subspace (the number of components) and the intervention method (e.g., scaling diagonally for 2D) appear a bit heuristic and need better justification.”
>
> We do have simple explanations for these:
>
> Intervention: We wanted to be able to plot the results in 3D without dimensionality reduction (Figure 5) because we thought it was very interesting to be able to do that. We chose to leave out 7.3 specifically, per Line 182: “changing 7.3 does not have a strong effect on its own” (see Figure 2). We entirely agree that this point is just too covert in the current draft, so we will raise this decision when introducing the interv. experiment
>
> Choice of scaling: We use z score thresholding on the distribution of composition scores (>4) to choose which components to scale. On all components this led to choosing 1D except for the duplicate token head which was 2D. This is why we used diagonal scaling. The graphs in Figures 6 and 7 show these components compared to others, and we believe it’s agreeable that these are also visually very clear outliers and the choice is justifiable.

---

> > ### Comment · Reviewer_Veos · 2024-08-11
> >
> > I appreciate your efforts in responding to the reviews. The responses were generally helpful and they address some of the key concerns I had with the initial draft. Some additional comments:
> >
> > - I think the additional experiment with larger models is a meaningful result and something that should definitely be included in the revision (either in the main text or in the appendix). One follow-up question is how this result relates to the paper’s claim about a “capacity limit” in Section 5. Am I correct in thinking that a way to address 10+ items is to increase the residual stream dimension to the point where the top-k inhibition components give me enough power to address all objects? Or is there something more fundamental about the task that cannot be addressed simply by increasing the model size? I’m still curious as to exactly what parts (if any) of your analyses and implications would be affected by having larger models.
> > - I think the baselines and the details about intervention experiments in your response should be included in the paper (could be referenced in an appendix).
> > - Regarding overclaims, I think your response covered some of my concerns but not all (I already listed 5 specific examples in my original review). To highlight one: I think the title should specify that the communication channels concern compositional operations (or just mention inhibition directly), or that the paper primarily discusses object identification/recall tasks, or anything that is more specific. This is clearly not the first paper to discuss *any* type of inter-layer communication in Transformer LMs.
> > - More generally, I still feel that the gap from “the inhibition-mover subcircuit for object identification/recall tasks” to “how information passed across layers in Transformers” is quite large, and that the paper should acknowledge this rather than giving the impression that the results generalize more broadly. We don’t even know what other types of channels may exist or whether we can still identify low-rank channels using the composition score for those. Again, this is not at all to say the paper’s study is uninteresting, but I just believe the general tone of the paper needs to be improved to focus more on the actual objects/tasks being studied. That said, enhancing the related work section and properly disambiguating from existing terminologies should alleviate some of the concerns.
> >
> > I have updated my ratings to reflect both the original draft and the authors’ rebuttal.

---

> > > ### Author Response · Authors · 2024-08-12
> > > **Thank you for the reply**
> > >
> > > We appreciate the reviewers effort in considering our (lengthy) rebuttal and are glad this had an effect on their perception of our work.
> > >
> > > > Am I correct in thinking that a way to address 10+ items is to increase the residual stream dimension to the point where the top-k inhibition components give me enough power to address all objects
> > >
> > > Yes, that is our intuition as well, for one, for one, because we see performance degrade more slowly as model size increases. But another reason is that we expect more inhibition-type components/heads to be present in larger models. This intuition comes from the finding of redundant computations in larger models (e.g., see [Lieberum et al. 2023](https://arxiv.org/abs/2307.09458)). With the additional time we will keep trying larger models until we seem to saturate the task.
> > >
> > > > To highlight one: I think the title should specify that the communication channels concern compositional operations (or just mention inhibition directly)
> > >
> > > We are understanding this point a lot more clearly now. We agree this is fair point. We'll think about specific titles more but we can change it to something like "...communication between layers in context recall tasks". We want to retain the key point that this is about layerwise movement rather than inter-residual stream movement through attention (more broadly studied)
> > >
> > > We will spare going into the every remaining detail, but the remaining related points are well taken and we will reframe where necessary. We will include an appendix showing that the decomposed composition score finds other very low rank outliers that exist, but we don't yet understand functionally (unrelated to the inhibition-mover subcircuit).

---

### Official Review · Reviewer_WKbg · 2024-07-16

**Soundness:** 2
**Presentation:** 3
**Contribution:** 2
**Rating:** 5
**Confidence:** 4

**Summary:**

This paper primarily employs low-rank communication channels to elucidate how internal layers within the Transformer model transmit information. Initially, the study utilizes existing research to identify duplicate heads, inhibition heads, and mover heads. The objective is then to explore the interactions among these heads. The findings indicate that directly incorporating these elements into the computation of the Composition Score does not yield significant results due to the complexity of the signals involved. Consequently, the paper proposes the application of Singular Value Decomposition (SVD) on the weight matrix, with subsequent sorting by variance, to identify the principal read/write subspace. This approach identifies such communication channels solely through the model's weight file. Furthermore, in certain recall tasks, such as the Indirect Object Identification (IOI) and Laundry List Task, this paper leverages the low-rank characteristics of communication channels to enhance task accuracy.

**Strengths:**

1. This article provides a detailed background to facilitate readers' understanding of prior research.
2. The article presents a hypothesis, verifies it through experiments, and conducts further in-depth research, thereby demonstrating a complete scientific research process.
3. To validate the proposed hypothesis, the article also develops a Laundry List Task for specific experimental verification.

**Weaknesses:**

1. As part of its contributions, this article presents a method for identifying nearly complete IOI circuit signals in GPT-2 Small directly through the weight file. However, the instructions provided in lines [148-152] are too brief, making the specific method difficult to understand.
2. It is well known that the matrix of the attention head is sparse, which naturally suggests using Singular Value Decomposition (SVD) to compress the original matrix. Although the primary aim of this article is to identify subspaces, there is no fundamental difference between the two approaches. Numerous related works, such as LoRA, have applied SVD to the original model weights. Moreover, the conclusion that inter-layer communication is low-rank seems evident due to the inherent low rank of the attention head weight matrix.

**Questions:**

1. The primary experiments in this paper were conducted on the IOI and Laundry List Task. Notably, the Laundry List Task has significant limitations in excluding repeated words. Inhibiting repeated words can only be considered a part of the recall task. Is it possible that recall tasks are the ones where repeated words appear most frequently? Some tasks should enhance the inhibitory effect on repeated words, while others should reduce it. Therefore, the experimental tasks in this paper are insufficient.
2. In Figure 2, the Inhibition Score is presented, but it is not introduced until Section 3.3.1, which may confuse readers. The use of \( V \) in Equation 2 may confuse readers with the \( V \) matrix in the Transformer model. There is a spelling error in the caption of Figure 3: "inhiibt" should be "inhibit."
3. The model-editing method proposed in this article involves performing SVD decomposition on a specific head (the inhibition head) and retaining the main subspace components. This approach improves performance on the Laundry List Task, indicating an enhanced inhibitory effect on repeated words. The intervention method in this article primarily demonstrates that other signals unrelated to inhibitory ability exist within the inhibition head, and that dimension reduction can enhance inhibitory capacity.  It should be noted that identifying the inhibition head is not a contribution of this article; rather, it relies on the conclusions of previous work. This article merely reinforces that the inhibition head functions as an inhibitory mechanism and that its combination with the duplicate head improves the inhibition of repeated words, a conclusion already established in prior research. Additionally, this article does not propose a new mechanism for inter-layer communication in the Transformer model.

**Limitations:**

The author has not fully addressed the limitations of their research. While the proposed method improves the model's inhibitory effect on repeated words, it primarily relies on existing findings and does not introduce a new mechanism for inter-layer communication in the Transformer model. Additionally, the experimental tasks used in the study, such as the Laundry List Task, have notable limitations, enhance the inhibitory effect on repeated words, while others maybe should reduce it.

---

> ### Author Rebuttal · Authors · 2024-08-07
>
> > “It is well known that the matrix of the attention head is sparse, which naturally suggests using Singular Value Decomposition (SVD) to compress the original matrix… the conclusion that inter-layer communication is low-rank seems evident due to the inherent low rank of the attention head weight matrix”
>
> Attention heads are inherently low rank because of the size of the matrices (e.g., 64x768, projecting down to 64 dims). This is much different from the low rank subspaces we identify here, which are 1d or 2d. We disagree with the claim that this is somehow self-evident, because we find that not all compositions between communicating matrices have this property. For example, see FIgure 6, left. We know that previous token head 4.11 composes with induction head 5.5 from prior circuit analysis (Wang et al., 2023), but they seem to do so with nearly the full-rank space of the attention head.
>
> > “Numerous related works, such as LoRA, have applied SVD to the original model weights.”
>
> Does the above point address the concern about the connection to LoRA? If not, could the reviewer please expand on this? We are familiar with some of the work on LoRA that is relevant but we are not aware of any work that implies the results we find in this paper.
>
> > “the instructions provided in lines [148-152] are too brief, making the specific method difficult to understand.”
>
> Thanks for raising this. We take the distribution of composition scores between component matrices and another matrix (such as mover head 9.9) and take the highest z score components (the outliers). We do not determine an optimal threshold in this work, but we use >4 (upon visual inspection the outliers are extremely obvious: see Figure 6 middle and right).
>
> > “In Figure 2, the Inhibition Score is presented, but it is not introduced until Section 3.3.1, which may confuse readers. The use of ( V ) in Equation 2 may confuse readers with the ( V ) matrix in the Transformer model. “
>
> Thanks for pointing this out, we will address this in the camera ready by introducing the term earlier.
>
> >  “This article merely reinforces that the inhibition head functions as an inhibitory mechanism”
> No other work has been able to find a known circuit within a model solely from the weights without running the model. We think this is a significant contribution to the field. We focus on a well known circuit to establish credibility without too much overhead of verifying a new circuit.
>
> We localize the inhibition mechanism to a few dimensions in the attention heads. We establish that traversing this space (by intervening on the outputs of these heads) controls the position of the token inhibited completely independently of the content of the token. That is, the edit we make is ‘ignorant’ to the preceding context, yet the downstream effect is predictable. This has not been established by previous literature
>
> > “this article does not propose a new mechanism for inter-layer communication in the Transformer model.”
>
> The new mechanism is the way that this is implemented in the model, with extremely low rank signals. ‘Bandwidth’ in the residual stream has been hypothesized as being in high demand (Elhage et al., 2021), but it has not been established how these signals are passed. We thoroughly justify the claim that this is one such method native to the transformer (see point 1).
>
> We believe we have clarified the points raised by the reviewer. This paper makes heavy use of the appendix for supporting figures, which we know reviewers are not necessarily responsible for, so we have provided some pointers here. We hope the reviewer considers these in their determination of the paper.

---

### Author Rebuttal · Authors · 2024-08-07

We would like to thank the reviewers for their thorough and thoughtful comments and suggestions. We are glad the reviewers shared our interest in the results on static weight analysis, as we’re hopeful for continued progress in this area, and we’re pleased that there was mostly consensus that are findings supported our hypotheses. After going over the reviews, the main concerns appear related to whether the main mechanism we discuss (related to inhibition) really matters outside of GPT2-small and pythia. We’d like to dedicate the rebuttal to addressing those concerns.

# Other Models
First, we’d like to briefly discuss and expand on the implications of our findings beyond the small models tested here. In Figure 1 of the rebuttal, we wanted to highlight that larger/more modern models do not solve the object selection problem we use to motivate the paper in Figure 1. This is to make the point that we did not pick a mechanism too specific to make broader contributions. We’d also like to draw attention to [Liu et al., 2023](https://arxiv.org/abs/2307.03172) which shows that even very powerful LMs can struggle with recall from context. More generally, we believe our modified composition score can beyond the models tested here. The concept of composition score (and its decomposed variant introduced here) generalizes to mlp-mlp and attn-mlp interactions as well, which we don’t test here. We also believe that the issue with positional embeddings and query/key composition can be solved in the future. Note that we do have positive results with pythia (which uses RoPE) and value composition in this paper. This paper is simply too full already to include such extensions to the work and are hoping to explore these directions in follow ups.

# Other Tasks:
There was some concern from a few reviewers that the tasks studied here (and therefore the mechanisms involved) are too narrow. We argue that the mechanism we identify is part of a more basic language modeling mechanism that underlies most tasks that involve recall from context. In Figure 2 of the rebuttal we show examples of the inhibition heads activity . Some reviewers pointed out that we were vague about our methods that led to claims in Lines 228-231, so we outline them below.
Combined with the observations about larger models above, we believe our results indicate that although we are targeting an extremely specific mechanism, this is actually part of a more general language model capability present in LMs. Thus, we see our findings and our methods of analysis useful and productive progress towards a greater understanding of the inner workings of LMs.

## Details on Identifying passages that highly activate inhibition heads
Some reviewers pointed out that we were not specific about how we arrived at the claims in Lines 228-231. We would like to provide our methodology which was quite rigorous. First, we split passages from OpenWebText-10K into token sequences of length 256 and run them through the model. We cache the attention patterns of the inhibition heads and manually examine tokens for which the attention score is >=0.3 for any token besides the very first. We use this to define “highly active”. We examined about 200 of these passages manually and found the patterns to be extremely consistent. The heads would almost always attend from some token like “and” to some other token that would most likely be a repetitive continuation, e.g., in “Crime and”, the “and” would attend to “Crime” presumably to inhibit “Crime and Crime” from being generated. Figure 2 left shows a screenshot of the interface we built to examine these examples when preparing this manuscript. Note that these were not cherry picked, they are the first examples from the dataset that arise from the .3 threshold. We'd like to add a section in the appendix for further analysis of inhibition heads on open data



# Other edits
We’d lastly like to thank the reviewers for their attention and specific feedback on the presentation of the results. This paper has many small details, and we appreciate the patience that was needed to be able to provide such low-level feedback. After implementing reviewer edits, we have some space left on page 9, which we will dedicate to discussion and expanding related work, which was suggested.

---

### Decision · Program_Chairs · 2024-09-25

**Decision:**

Accept (poster)

**Comment:**

This paper identifies particular mechanisms by which information is communicated between transformer layers. They relate their identified "head type" mechanisms to a particular simple task, and use this to conduct experiments demonstrating the (interventional) effects of targeted modifications following their decomposition.

Reviewers generally agree that the results are interesting and worth publishing. Concerns focused on the degree to which the paper over claims, the clarity of presentation, and whether and to what extent the model and experiments here really have general predictive power for LLMs. (See particularly the discussion with Reviewer Veos).